



# Derivation of the Reduced Reaction Mechanisms of Ozone Depletion Events in Polar Spring by Using Concentration Sensitivity Analysis and Principal Component Analysis

Le Cao[1], Chenggang Wang[1], Mao Mao[1], Holger Grosshans[2], and Nianwen Cao[1]

[1]Key Laboratory for Aerosol-Cloud-Precipitation of China Meteorological Administration, Nanjing University of Information Science and Technology, Nanjing, China
[2]Institute of Mechanics, Materials and Civil Engineering, Université catholique de Louvain, Louvain-la-Neuve, Belgium

*Correspondence to:* L. Cao
(le.cao@nuist.edu.cn)

**Abstract.** The ozone depletion events (ODEs) in the spring of Arctic have been investigated since the 1980s. It is found that the depletion of ozone is highly associated with an auto-catalytic reaction cycle in which the bromine containing compounds are mostly involved. Moreover, bromide stored in various substrates in the Arctic such as the underlying surface covered by ice and snow can be also activated by the absorbed HOBr. Subsequently this leads to an explosive increase of the bromine amount
in the troposphere, which is the so-called "bromine explosion mechanism".

In the present study, a reaction scheme representing the chemistry of ozone depletion and halogen release is processed with two different mechanism reduction approaches, namely the concentration sensitivity analysis and the principal component analysis. In the concentration sensitivity analysis, the interdependence of the mixing ratios of ozone and principal bromine species on the rate of each reaction in the mechanism of ODEs is identified. Furthermore, the most influential reactions in
different time periods of ODEs are also revealed. By removing 11 reactions with the maximum absolute values of sensitivities lower than 10%, a reduced reaction mechanism of ODEs is derived. The onsets of each time period of ODEs in simulations using the original reaction mechanism and the reduced reaction mechanism are identical while the maximum deviation of the mixing ratio of principal bromine species between different mechanisms is found less than 1%.

By performing the principal component analysis on an array of the sensitivity matrices, the dependence of a particular
species concentration on a combination of the reaction rates in the mechanism is revealed. Redundant reactions are indicated by principal components corresponding to small eigenvalues and insignificant elements in principal components with large eigenvalues. Through this investigation, aside from the 11 reactions which have been identified as unimportant in the concentration sensitivity analysis, additionally nine reactions were identified to contribute only little to the total response of the system. Thus, they can be eliminated from the original reaction scheme. The results computed by applying the reduced reaction
mechanism derived after the principal component analysis agree well with those by using the original reaction scheme. The maximum deviation of the mixing ratio of principal bromine species is found less than 10%, which is guaranteed by the selection criterion adopted in the simplification process. Moreover, it is shown in the principal component analysis that O($^1$D) in





the mechanism of ODEs is in quasi-steady state, which enables a following simplification of the reduced reaction mechanism obtained in the present study.

## 1 Introduction

Since the first discovery by Oltmans (1981) in observations conducted at Barrow, Alaska in 1977, the ozone depletion events

(ODEs) in polar spring have attracted great attention. It was found by Oltmans (1981) that the surface ozone mixing ratio in Barrow dropped from a background value ($\sim$40 ppb, ppb = parts per billion) to a level lower than 1 ppb within a couple of days. In a following measurement performed at Alert, Canada, not only the occurrence of the ozone depletion but also a negative correlation between the ozone mixing ratio and the concentration of filterable bromine (f-Br) was confirmed (Bottenheim et al., 1986; Barrie et al., 1988). Since then, ODEs and the associated bromine enhancement have been reported from various

observation sites in polar regions (Kreher et al., 1997; Frieß et al., 2004; Jones et al., 2006; Wagner et al., 2007; Helmig et al., 2007; Jones et al., 2009, 2010; Helmig et al., 2012; Halfacre et al., 2013).

Although, the concentration increase of the brominated species was found during ODEs, the mechanism responsible for the destruction of ozone with the involvement of bromine in the troposphere of Arctic remained a matter of debate until the role of bromine monoxide (BrO) was uncovered by Hausmann and Platt (1994). It was suggested by Hausmann and Platt (1994) that

the bromine species participate in an auto-catalytic reaction cycle (I), namely

$$2\,(\mathrm{Br} + \mathrm{O}_3 \rightarrow \mathrm{BrO} + \mathrm{O}_2)$$
$$\mathrm{BrO} + \mathrm{BrO} \rightarrow \mathrm{Br}_2\,(\mathrm{or}\,2\,\mathrm{Br}) + \mathrm{O}_2$$
$$\underline{\mathrm{Br}_2 + h\nu \rightarrow 2\,\mathrm{Br}}$$
$$\mathrm{Net}: 2\,\mathrm{O}_3 + h\nu \rightarrow 3\,\mathrm{O}_2 \tag{I}$$

through which the ozone in the boundary layer is consumed without any loss of bromine. In the reaction sequence (I), BrO participates in self reactions through which Br atoms are formed. Since the reaction between ozone and Br is rapid, ozone in the boundary layer is continuously consumed in the presence of sunlight via reaction sequence (I). However, the reaction cycle

(I) is unable to explain the fast enhancement of bromine in the ambient air. Therefore, another reaction cycle (II) is proposed as follows,

$$\mathrm{Br} + \mathrm{O}_3 \rightarrow \mathrm{BrO} + \mathrm{O}_2$$
$$\mathrm{BrO} + \mathrm{HO}_2 \rightarrow \mathrm{HOBr} + \mathrm{O}_2$$
$$\mathrm{HOBr} + \mathrm{H}^+ + \mathrm{Br}^- \overset{\mathrm{mp}}{\Rightarrow} \mathrm{Br}_2 + \mathrm{H}_2\mathrm{O}$$
$$\underline{\mathrm{Br}_2 + h\nu \rightarrow 2\,\mathrm{Br}}$$
$$\mathrm{Net}: \mathrm{O}_3 + \mathrm{HO}_2 + \mathrm{H}^+ + \mathrm{Br}^- + h\nu \overset{\mathrm{mp}}{\Rightarrow} 2\,\mathrm{O}_2 + \mathrm{Br} + \mathrm{H}_2\mathrm{O}. \tag{II}$$

In sequence (II), the HOBr, which is formed through the oxidation of BrO, is capable of activating bromide ions stored in various polar substrates such as the suspended aerosols and ice/snow-covered surfaces. As a result, the bromine amount in





the ambient air rises up explosively. This reaction sequence is thus named as "bromine explosion mechanism" (Platt and Janssen, 1995; Platt and Lehrer, 1997; Wennberg, 1999). A detailed review of the auto-catalytic reaction cycles of ODEs and the bromine explosion mechanism is given by Platt and Hönninger (2003), Simpson et al. (2007) and Abbatt et al. (2012).

Since the 1990s, in order to reveal the physical and chemical processes associated with ODEs, a large amount of numerical studies on the ozone depletion and halogen release in polar spring have been conducted. Box models, also termed 0-D models, were first used, in which the chemical species are assumed to be distributed uniformly within the boundary layer. The vertical mixing is usually parameterized in box models to estimate the turbulent mixing intensity in the boundary layer. By using a box model, McConnell et al. (1992) first suggested that the heterogeneous reactions which release Br⁻ from the snowpack to the atmosphere are critical for sustaining high levels of BrO and Br in the ambient air so that ozone can be consumed rapidly. However, they did not specify the form of the heterogeneous reactions. Afterwards, based on a known aqueous-phase chemistry, Fan and Jacob (1992) proposed a form of the mechanism for the release of $Br_2$ from the suspended aerosols. Their model results show that the production of $Br_2$ in the aerosols is an essential factor for the occurrence of ODEs.

In the year 1996, Tang and McConnell used a box model to investigate the source of bromine at polar sunrise, and proposed that the snowpack above the sea ice surface is a possible bromine source in the Arctic. The importance of the snowpack during the Arctic spring ozone depletion was also identified by Michalowski et al. (2000) by using the multiphase box model CREAMS (Chemical Reactions Modeling System). They elaborated that the ice surface or snowpack is possibly a major reservoir for the heterogeneous halogen recycling. Moreover, they claimed that the mass transfer rate of ozone and halogen acids between the snowpack and the ambient air is the determining factor to the ozone depletion. In 2003, Evans et al. used an ozone coordinate to compare the results obtained from a photochemical box model with the observations during the TOPSE (Tropospheric Ozone Production around the Spring Equinox) campaign. The processes controlling the coupled evolution of $BrO_x$-$ClO_x$-$HO_x$-$NO_x$ chemistry during ODEs were also investigated. Based on the measurements of $HO_x$, $NO_x$, halogen containing compounds and ozone during the CHABLIS (Chemistry of the Antarctic Boundary Layer and the Interface with Snow) campaign (Jones et al., 2008), Bloss et al. (2010) developed a box model by adding a halogen reaction scheme to the standard chemical reaction mechanism MCM (Master Chemical Mechanism, http://mcm.leeds.ac.uk/). The diurnal profiles of $HO_x$ and $NO_x$ are captured in the model, and the influences of halogen species on the mixing ratios of $HO_x$ and $NO_x$ are investigated as well. In 2014, the author of the present manuscript and co-workers (Cao et al., 2014) worked on the box model KINAL (Turányi, 1990a) to perform a concentration sensitivity analysis on a reaction mechanism of ODEs. By doing so, they revealed a series of important reactions which determine the occurrence of ODEs.

The photochemical box model MOCCA (Model Of Chemistry in Clouds and Aerosols) was developed by Sander and Crutzen (1996), initially for the aim of investigating the role of bromine in the polluted air advected over the marine boundary layer. Different from Fan and Jacob's model mentioned above, in MOCCA, the aerosol particle behaves as an additional source of bromine with the assumption that the bromine ions in the aerosols are inexhaustible. In a following work by Sander et al. (1997), MOCCA was modified to adapt to polar conditions, which leads to a new model named MoccaIce (Model Of Chemistry Considering Aerosols In Cold Environments). By using MoccaIce, the chemistry of ozone depletion and bromine explosion in the troposphere during Arctic spring was investigated. It was found that the rates of reactions between Br and $C_2H_2$ or



$C_2H_4$ are critical for the ozone budget in the troposphere. Later on, a new comprehensive atmospheric chemistry box model MECCA (Module Efficiently Calculating the Chemistry of the Atmosphere) was presented by Sander et al. (2005). The results of MOCCA (Sander and Crutzen, 1996) were reproduced by MECCA, and a comparison between these two box models was conducted. Furthermore, MECCA was used by Sander et al. (2006) to reveal the triggering of the bromine explosion and ozone

destruction mechanism. It was proposed by Sander et al. (2006) that the precipitation of $CaCO_3$ from the freezing sea water is a key factor of ODEs as it reduces the buffering ability of the sea water, which facilitates the acid-catalyzed activation process.

However, by utilizing the equilibrium thermodynamic model FREZCHEM, Morin et al. (2008) indicated the risk of using the results from Sander et al. (2006). Based on the finding of Dieckmann et al. (2008) that ikaite ($CaCO_3 \cdot 6\,H_2O$) precipitates in brine instead of calcite ($CaCO_3$), Morin et al. (2008) suggested that if calcite is the major calcium carbonate mineral that

precipitates, the results from Sander et al. (2006) are valid. On the contrary, if ikaite precipitates during the brine freezing instead of calcite, the alkalinity of aerosol would not be buffered, which contradicts the conclusion made by Sander et al. (2006). Later on, in a joint publication of Sander and Morin (2010), the authors showed by re-analyzing the results from MECCA and FREZCHEM that the conclusions of Sander et al. (2006) hold as long as the mechanism of evapoconcentration during the formation of aerosol particles is considered. In addition, they also suggested that the conclusions of Sander et al.

(2006) and Morin et al. (2008) can be reconciled by the introduction of a bromide/alkalinity ratio.

The one-dimensional model studies of ODEs started from Lehrer et al. (2004) by using FACSIMILE to inquire the relative importance of sea salt aerosols and the fresh sea ice surface for the ozone destruction. In their 1-D model, the vertical distribution of the turbulent diffusivity is parameterized using a linear function of height. It was indicated by their model that the primary bromine source during ODEs is possibly the fresh sea ice surface which provides a huge amount of reactive surface

and nearly inexhaustible bromine compared to the sea salt aerosols. The prerequisites for the occurrence of ODEs were also discussed by Lehrer et al. (2004). Another 1-D chemical transport model, THAMO (Tropospheric HAlogen chemistry MOdel), was developed by Saiz-Lopez et al. (2008) with the aim of clarifying the source strengths of bromine and iodine required for maintaining the vertical profiles of BrO and IO observed during the CHABLIS field campaign. In their model, prescribed $Br_2$ and atom I fluxes from the snowpack are added, and the heterogeneous reprocessing of bromine and iodine on sea salt

aerosols is also included. It was found that an initial $Br_2$ flux is required for keeping the model simulations in accordance with the DOAS measurements of the CHABLIS campaign. The recycling of bromine on sea salt aerosols was also found essential for yielding a uniform BrO vertical distribution. Recently, Cao et al. (2016a) added a module representing the vertical mass transfer between layers with different heights to KINAL, and the new 1-D model is referred to as KINAL-T. Utilizing this 1-D model, the influences caused by the change of the boundary layer height on the occurrence and termination of ODEs were

studied.

A comprehensive model study of ODEs has been conducted by using the 1-D Lagrangian-mode model MISTRA (MIcrophysical STRAtus) (Bott et al., 1996; Bott, 1997, 2000). MISTRA is a boundary layer model initially developed for the investigation of the cloud-topped planetary boundary layer. Afterwards, the gas-phase and aqueous-phase reactions were implemented in MISTRA (von Glasow et al., 2002a, b; von Glasow and Crutzen, 2004). The new model is named as MISTRA-MPIC

(MISTRA Max-Planck-Institut für Chemie) and used for the study of the marine boundary layer at mid-latitude. The halogen





chemistry in a marine boundary layer is modeled in MISTRA-MPIC, and the vertical structure of chemical species such as BrO in a cloud-free marine boundary layer is also captured. MISTRA was then modified by Piot and von Glasow (2008) to reproduce Arctic conditions for the investigation of the role of frost-flower derived aerosols, open leads and re-release processes on the snowpack in the destruction of ozone during ODEs. It was found that the recycling process on snow is the most important

process for the existence of high-level bromine observed in the polar boundary layer. The effects brought about by the change of meteorological parameters such as the temperature of the ambient air were also studied.

In another publication of Piot and von Glasow (2009), the MISTRA model was switched to the box model mode for evaluating the influence of flux change of HCHO, $NO_x$, and $Cl_2$ on the ozone depletion rate. It was found that the elevation of HCHO and $Cl_2$ fluxes leads to a bromine speciation shift from Br/BrO to HOBr/HBr, which accelerates the deposition of bromine on

snow. As a result, the total reactive bromine remaining in the air is reduced, which slows down ODEs. Afterwards, Thomas et al. (2011, 2012) coupled a snowpack module to MISTRA, and the new physicochemical system is referred to as MISTRA-SNOW. The uptake, recycling and release of bromine in the interstitial air of snowpack and on snow grains were treated in great detail. The model successfully reproduces the diurnal behavior of trace gases such as BrO found in the observations during the GSHOX (Greenland Summit Halogen-$HO_x$) experiment. The vertical distributions of trace gases in the interstitial

air of snowpack as well as in the boundary layer are also well captured in MISTRA-SNOW. After a three-day model run, it was found that significant bromide is still available in the liquid-like layer of the snowpack, which indicates that the snowpack is able to provide adequate reactive bromine to sustain the BrO level observed during the GSHOX experiment.

In order to address the influence of snowpack on the ozone loss in the polar boundary layer, Toyota et al. (2014) developed the 1-D model PHANTAS (PHotochemistry ANd Transport between Air and Snowpack). Therein, great attention was paid

to the model configuration in order to realistically simulate the exchange between the snow interstitial air and the ambient air in the boundary layer. Both the rapid ozone depletion in the boundary layer and the enhanced bromine in the interstitial air were successfully simulated applying PHANTAS. It was also found that in the top layers of the snowpack, the release of $Br_2$ is dominantly driven by the bromine explosion mechanism, whilst in deeper layers of snowpack, both the aqueous radical chemistry and the bromine explosion mechanism contribute substantially to the bromine evasion to the boundary layer.

The three-dimensional model studies on the tropospheric ODEs during polar spring started from Zeng et al. (2003). In their 3-D regional chemistry transport model (RCTM), fed with the data of BrO obtained from the spectral measurements aboard GOME, low ozone episodes occurring at northern high latitude during the TOPSE campaign were successfully captured. Moreover, the correlation coefficients between the observed and modelled ozone temporal variations at different sites such as Alert and Barrow were found to be above 0.5. It was suggested by Zeng et al. (2003) that ODEs persist widespread in a

large scale. Later on, Zeng et al. (2006) modified the closure scheme of the meteorology field in their model. It was found that the decrease of the minimum value of the diffusion coefficient gives a better agreement of the ozone variations between the observations and the model results. In the year 2005, Yang et al. added a detailed bromine chemistry scheme to a global 3-D tropospheric model p-TOMCAT. Subsequently, they conducted seven simulations with different initial conditions to illustrate the contribution of each bromine source to the total bromine amount in the air. The lifetime and the vertical profile of BrO were

also investigated. In 2010, Yang et al. also added blowing snow as an additional bromine source to p-TOMCAT. Moreover,



the heterogeneous reactions responsible for the reactivation of inactive bromine species in aerosols were implemented as well. It was found that the bromine explosion cannot be predicted if the blowing snow module is switched off, which indicates the importance of blowing snow for the bromine explosion.

A major advancement of the 3-D models for studying bromine explosion and ozone consumption in the Arctic was made by
Zhao et al. (2008). They used a global 3-D chemistry and transport model GEM-AQ/Arctic to investigate the spatial structure and time series of ozone and BrO in the boundary layer during the Arctic springs of the years 2000 and 2001. In their model, the chemistry and aerosol modules are online coupled with the meteorology model. The chemistry module MECCA (Sander et al., 2005, 2006) described above was also implemented in GEM-AQ/Arctic, replacing the original chemistry module and solver. Moreover, the precipitation of $CaCO_3$ for triggering the bromine explosion outlined by Sander et al. (2006) is also
included in their global model. A good agreement was achieved between the GEM-AQ/Arctic simulations and the satellite observations. Additionally, it was shown in their model results that besides the halogen chemistry also the air temperature, atmospheric circulation, and long-range transport of pollutants make great contributions to the springtime ozone depletion in Arctic.

Another version of the 3-D model GEM-AQ with the incorporation of the gas-phase and heterogeneous bromine chemistry
was employed by Toyota et al. (2011) to discover the surface sources of bromine for the occurrence of ODEs and BrO-clouds observed in the boundary layer of the Arctic (Chance, 1998; Richter et al., 1998; Platt and Wagner, 1998; Wagner et al., 2001). A source representing the oxidation of Br$^-$ by ozone at the snow/ice-covered surfaces was included which later proved to be the primary source of reactive bromine. The simulation results compare reasonably well with ground-based measurements of the temporal behavior of surface ozone. The vertical distribution of ozone obtained by the model is also in consistence with the
ozonesonde data measured at different locations (Alert, Resolute and Ny Ålesund). In 2013, Cao and Gutheil coupled the 3-D model to a LES (Large Eddy Simulation) of the atmospheric flow using OpenFOAM in order to gain a better understanding of the turbulent mixing of ozone and bromine related species in the boundary layer during ODEs. In their paper they also discuss the dependence of the ozone depletion rate on meteorological conditions such as the wind speed and the boundary layer stability.

The above discussed theoretical studies revealed that ODEs are influenced by the joint effect of horizontal advection, vertical convection, and local chemistry. The potential to provide the information which may enable to fully understand the underlying physicochemical processes is offered by the numerical simulation of ODEs of high dimensions. However, the possibility of conducting high-dimensional computations is usually limited by the effort to calculate the chemical production and destruction source terms in the governing equations. If a complex reaction mechanism is adopted in this kind of simulations, the execution
time for the estimation of chemical source terms can exceed the one for solving transport equations by one order of magnitude.

Commonly, a one-step global reaction is adopted in 2-D or 3-D simulations of the atmospheric chemistry which leads to affordable computing times (Warnatz, 1992). However, the coefficients used in the one-step global reaction approach are usually empirical which cannot be estimated without the aid of experimental studies. An alternative treatment of the chemistry in numerical studies is to choose a couple of elementary reactions which are able to describe the important features of the
reaction system. However, the selection of elementary reactions needs special attention. If the treatment of the chemistry in





the computation is too rough and arbitrary, the estimation of the chemical production and consumption would be inaccurate and, consequently, the simulation results may be far off from reality. Thus, an appropriate reaction mechanism with reduced size and adequate accuracy is required so that the multi-dimensional simulations of ODEs are applicable. The adopted reaction mechanism is required to maintain the major properties of the original complex reaction mechanism. Moreover, the size of the

reaction mechanism should be small to enable the efficient computation of high-dimensional equations.

Previously, various approaches have been proposed to simplify a complex reaction mechanism while the major properties of the original reaction mechanism are maintained. The first type of the methods is represented by the rate spectrum analysis of the reaction scheme (Turányi et al., 1989). Therein, the rates of competing reactions are compared. However, this method is inapplicable for a very complex chemical system in which the influence of each reaction differs significantly on certain

kinetic features of the system. The second type of methods is given by the grouping of the reactions into several categories (Edelson and Allara, 1980). The reaction which makes negligible contribution to the flux within its category is considered to be unimportant, and, thus, can be eliminated from the reaction scheme. However, the grouping procedure of the reactions depends strongly on the experience of the investigator, which brings some arbitrariness to the application of this method. Moreover, the kinetic information inherent in the reaction mechanism may be lost if reactions belonging to a certain chemical kinetic

structure are grouped separately. The third method is based on the analysis of the production rate of free radicals (Lifshitz and Frenklach, 1975). Reactions which make small contributions to the production or destruction of free radicals are omitted. The fourth approach for obtaining the reduced mechanism is referred to as rate-of-production analysis or reaction path analysis (Glarborg et al., 1986). Herein, the individual contribution of each reaction to the overall production rate of selected chemical species is estimated. Also, the major formation pathway of the chemical species is identified.

A more complicated method which is able to extract the inherent information from the kinetic mechanism and discriminate reactions from complex chemical reaction schemes is the sensitivity analysis (Rabitz et al., 1983; Turányi, 1990b). Sensitivity analysis is a routinely-used technique which reveals interdependence between the species concentrations and the change of reaction rates. Finally, it helps to reduce the size of the reaction mechanism. At present, the sensitivity analysis is often conducted in a "brute force" way (Dodge and Hecht, 1975; Valko and Vajda, 1984) by simply applying a perturbation of reaction rate

coefficients by a fixed amount to see the corresponding deviations of particular species concentrations. This is time-consuming and provides only a limited accuracy. Since the 1980s, more systematic and economical approaches for the estimation of the sensitivities have been proposed. The most straightforward one is the concentration sensitivity analysis (Turányi, 1990b), which is capable of displaying the relative importance of an individual rate coefficient on a group of species concentrations and identifying the rate-determining steps in the reaction scheme. The concentration sensitivity analysis has also been used for screening

reactions from a complicated reaction mechanism in various fields such as the investigation of combustion (Dougherty and Rabitz, 1980; Warnatz et al., 2001) and atmospheric chemistry (Turányi and Bérces, 1990).

However, the calculation of the concentration sensitivity coefficients depends on the time interval used in the model, which denotes that the concentration sensitivity is inherently time-depending. Thus, the accuracy of the sensitivity estimation is associated with the length of the time interval. Aside from this, the concentration sensitivity analysis only discovers the importance

of an individual input parameter of the chemical kinetic system for several species concentrations while usually the concentra-




tions are influenced by a group of parameters. Thus, another mechanism reduction approach, namely the principal component analysis, is suggested (Vajda et al., 1985). The principal component analysis is performed based on the calculation of eigenvalues and eigenvectors of the matrix $\widetilde{\boldsymbol{S}}^{\mathrm{T}}\widetilde{\boldsymbol{S}}$, wherein $\widetilde{\boldsymbol{S}}$ denotes an array of the relative concentration sensitivity matrices. Strongly interacting reactions which significantly influence the concentration of particular species are identified by principal

components with large eigenvalues. Moreover, further mechanistic details of the chemical kinetic system such as the species in quasi-steady state can also be provided by the principal component analysis.

Although the mechanism reduction techniques mentioned above have been applied in the investigation of various atmospheric phenomena such as the chemistry in clouds (Pandis and Seinfeld, 1989), to date, little effort has been paid to the reduction of the reaction mechanism of ODEs using these techniques. Previous research of the author and co-workers (Cao and

10 Gutheil, 2013; Cao et al., 2014) focused on this aspect to some extent. In these studies, relative concentration sensitivities of a reaction mechanism of ODEs were computed, and the simplification of the mechanism based on the concentration sensitivities was prospected. However, the details of the reduction processes were not presented. Besides, the reduction technique discussed in these studies (Cao and Gutheil, 2013; Cao et al., 2014) is only limited to the concentration sensitivity analysis. Thus, in the present study, two different approaches, concentration sensitivity analysis and principal component analysis, were applied

for reducing the size of a complex reaction mechanism of ozone depletion and halogen release. The results obtained by using these two methods are compared and discussed. The connection between different chemical reactions is also revealed during the reduction processes.

The manuscript is organized as follows. In Sect. 2, the governing equations of the reduction approaches used in this study are presented. The criteria for screening unimportant reactions from the original reaction scheme are also given in this section.

Then the computational results of this work are shown in Sect. 3. Reduced reaction mechanisms derived after the analyses are presented and compared as well. The species in quasi-steady state is also revealed by the implementation of the principal component analysis. At last, major conclusions made in the present study are addressed in Sect. 4. Further simplification of the reduced reaction mechanism is also prospected.

## 2 Mathematical Model and Methods

A complex chemical reaction system can be denoted as

$$\frac{\mathrm{d}\boldsymbol{c}}{\mathrm{d}t} = \boldsymbol{f}(\boldsymbol{c}, \boldsymbol{k}) + \boldsymbol{E}, \tag{1}$$

with the initial condition $\boldsymbol{c}|_{t=0} = \boldsymbol{c}_0$, where $\boldsymbol{c}$ is a column vector of species concentrations. $\boldsymbol{k}$ in Eq. (1) is a vector of reaction rate coefficients and $t$ denotes time. $\boldsymbol{E}$ represents the source term of local surface emissions. In the present study, a reaction mechanism with the involvement of bromine containing compounds and nitrogen related species is adopted from the previous

box model study (Cao et al., 2014). Furthermore, the reaction rate constants are updated with the latest chemical kinetic data (Atkinson et al., 2006). Several photolysis reactions are also added to complete the previous reaction mechanism. The reaction scheme used in the present study consists of 39 chemical species and 92 reactions and is listed in the supplementary material.



In this chemical reaction mechanism, bromine is considered as the only halogen species while the chlorine related species and reactions are excluded. The influences brought about by the inclusion of chlorine related species and reactions have been discussed by Cao et al. (2014). As the focus of the present study is laid on the reduction of the reaction mechanism instead of pursuing a more precise prediction of the temporal change of each chemical species, the choice of this chlorine-free reaction mechanism does not affect the metrics of the present study.

As mentioned above, two different approaches, concentration sensitivity analysis and principal component analysis, are followed to reduce the size of the reaction mechanism of ODEs. The governing equations of these two approaches and the procedures of the mechanism simplification are presented below.

## 2.1 Concentration Sensitivity Analysis

The importance of the $j$-th reaction for the $i$-th chemical species is depicted by the relative concentration sensitivity $\widetilde{S}_{ij}$, which can be written in the form of

$$\widetilde{S}_{ij} = \frac{\partial \ln c_i}{\partial \ln k_j} = \frac{k_j}{c_i}\frac{\partial c_i}{\partial k_j} = \frac{k_j}{c_i}S_{ij}. \tag{2}$$

In Eq. (2), $c_i$ is the concentration of the $i$-th chemical species, and $k_j$ denotes the rate coefficient of the $j$-th reaction. $S_{ij} = \partial c_i/\partial k_j$ is the absolute concentration sensitivity, and the unit of $S_{ij}$ depends on the order of the $j$-th reaction. In order to compare the sensitivity coefficients belonging to different reactions, the normalized version of the sensitivity coefficient, $\widetilde{S}_{ij}$, is introduced by multiplying $S_{ij}$ with $k_j/c_i$. The obtained relative concentration sensitivity $\widetilde{S}_{ij}$ is thus a dimensionless variable which represents the percentage of change in the $i$-th species concentration due to a 1% change of the $j$-th reaction rate constant. The evaluation of the relative concentration sensitivity is helpful for discovering the interdependence between the solution of Eq. (1) and the input parameters such as the reaction rate constants $k_j$.

The calculation of $\widetilde{S}_{ij}$ is performed by differentiating the $i$-th component of Eq. (1) with respect to $k_j$. With the assumption that the local emissions are independent of the rate constants, Eq. (1) becomes

$$\frac{\mathrm{d}(\partial c_i/\partial k_j)}{\mathrm{d}t} = \sum_{l=1}^{ns}\frac{\partial f_i}{\partial c_l}\frac{\partial c_l}{\partial k_j} + \frac{\partial f_i}{\partial k_j}. \tag{3}$$

The upper limit of the sum in Eq. (3), $ns$, denotes the total number of the chemical species included in the model. By substituting $\partial c_i/\partial k_j$ and $\partial c_l/\partial k_j$ in Eq. (3) with the absolute concentration sensitivities $S_{ij}$ and $S_{lj}$, we obtain another form of Eq. (3) as follows,

$$\frac{\mathrm{d}S_{ij}}{\mathrm{d}t} = \sum_{l=1}^{ns}\frac{\partial f_i}{\partial c_l}S_{lj} + \frac{\partial f_i}{\partial k_j}. \tag{4}$$

The second term of the right hand side of Eq. (4) denotes the direct influence on the concentration of $i$-th species caused by the change of the $j$-th reaction rate constant. Moreover, as a result of this direct variation in the $j$-th reaction rate, indirect effects on the concentrations of other species are induced via the coupled kinetic system, which contributes to the solution of Eq. (4). This indirect effect of parameter change is indicated by the first term of the right hand side of Eq. (4). After obtaining the





solution $S_{ij}$ of Eq. (4), the relative concentration sensitivity can be calculated by multiplying $S_{ij}$ with $k_j/c_i$. The computation of the absolute concentration sensitivity is conducted by using the subroutine SENS in the chemical kinetic software KINAL (Turányi, 1990a). The Decomposed Direct Method (Valko and Vajda, 1984) is implemented in KINAL for solving Eq. (4), which has been proved robust and efficient (Turányi, 1990b).

5    The concentration sensitivity analysis is a useful measure of how sensitive a specified species concentration is to a particular reaction rate constant. Thus, reactions with large absolute values of sensitivities are identified as important and rate-determining. For the purpose of deriving an appropriate reduced reaction mechanism of ODEs, it is needed to remove the least important reactions from the original reaction scheme. In the present study, we consider the $j$-th reaction as unimportant if the criterion

$$\max |\widetilde{S}_{ij}(t)| \leq 10\%; i = 1, ..., ns, t = t(1), ..., t(nt) \tag{5}$$

is fulfilled. In Eq. (5), $ns$ is the total number of the chemical species as mentioned above, and $nt$ denotes the total time step in the computation. The criterion shown in Eq. (5) means that the $j$-th reaction is considered as unimportant if its relative concentration sensitivity for all species at any time point is smaller than 10% and then can be eliminated from the system.

Although the concentration sensitivity analysis is powerful for constructing a minimal chemical reaction set describing ODEs, it has some disadvantages. As the concentration sensitivity represents the response of a system to a perturbation of the reaction rate at an earlier time point, the concentration sensitivity coefficient is inherently time-depending, and its estimation depends on the adopted time interval. Thus, the concentration sensitivity obtained by solving Eq. (4) corresponds to a time interval instead of a fixed time point. A drawback brought about by this inherent time-depending property of the concentration sensitivity is the "memory effect" (Vajda and Turányi, 1986; Turányi et al., 1989). Within the time interval for the integration of sensitivity coefficients, if a reaction is indicated as important at any stage of the time interval, it is identified as important during the whole time interval since this property is maintained in the concentration sensitivity analysis. As a result, this reaction cannot be eliminated from the original reaction scheme. In addition, although the concentration sensitivity analysis is capable of clarifying the dependence of species concentrations upon a particular rate coefficient, usually the species concentrations are influenced by a combination of input parameters such as a ratio of two reaction rate constants. This interdependence between the species concentrations and the groups of parameters can hardly be revealed by the concentration sensitivity analysis. Therefore, in the present study, another mechanism reduction approach is chosen, namely the principal component analysis, and applied to the original reaction mechanism of ODEs, which is discussed in the section below.

## 2.2    Principal Component Analysis

The principal component analysis provides an effective means of screening unimportant reactions from a complex reaction scheme so that a tractable reaction mechanism can be derived. To perform the principal component analysis, we first introduce the normalized rate parameter, $\boldsymbol{\alpha}$, of which the $j$-th component can be expressed as

$$\alpha_j = \ln k_j, j = 1, ..., np. \tag{6}$$





Herein, $np$ denotes the total number of the chemical reactions considered in the model.

The definition given by Eq. (6) enables to express the response of the reaction mechanism due to a variation in the rate parameters, $\boldsymbol{\alpha}$, using the response function $Q(\boldsymbol{\alpha})$ as follows,

$$Q(\boldsymbol{\alpha}) = \sum_{m=1}^{nt} \sum_{i=1}^{ns} \left[ \frac{c_{i,m}(\boldsymbol{\alpha}) - c_{i,m}(\boldsymbol{\alpha}^0)}{c_{i,m}(\boldsymbol{\alpha}^0)} \right]^2. \tag{7}$$

In Eq. (7), the subscript "$i,m$" represents the concentration of $i$-th species at the $m$-th time point. $ns$ and $nt$ are the total number of the chemical species and the time step, respectively. $\boldsymbol{\alpha}^0$ denotes a matrix of the normalized parameters with original values.

By applying a Taylor series expansion (Vajda et al., 1985) and a Gauss approximation (Bard, 1974) on Eq. (7), the response function $Q_{(}\boldsymbol{\alpha})$ is approximated as

$$Q(\boldsymbol{\alpha}) \approx \widetilde{Q}(\boldsymbol{\alpha}) = (\Delta\boldsymbol{\alpha})^{\mathrm{T}} \widetilde{\boldsymbol{S}}^{\mathrm{T}} \widetilde{\boldsymbol{S}} (\Delta\boldsymbol{\alpha}). \tag{8}$$

Herein, $\widetilde{Q}(\boldsymbol{\alpha})$ is an approximate response function. $\Delta\boldsymbol{\alpha}$ denotes a column vector of the parameter variations which has the $j$-th component,

$$\Delta\alpha_j = \ln\alpha_j - \ln\alpha_j^0. \tag{9}$$

$\widetilde{\boldsymbol{S}}$ in Eq. (8) represents an array of the relative concentration sensitivities corresponding to the time instance $t(m)$, $m = $
$1, \ldots, nt$, and the $m$-th element of $\widetilde{\boldsymbol{S}}$ has the form of

$$\widetilde{S}_m = \begin{pmatrix} \frac{\partial \ln c_{1,m}}{\partial \ln k_1} & \frac{\partial \ln c_{1,m}}{\partial \ln k_2} & \cdots & \frac{\partial \ln c_{1,m}}{\partial \ln k_{np}} \\ \frac{\partial \ln c_{2,m}}{\partial \ln k_1} & \frac{\partial \ln c_{2,m}}{\partial \ln k_2} & \cdots & \frac{\partial \ln c_{2,m}}{\partial \ln k_{np}} \\ \vdots & \vdots & \ddots & \vdots \\ \frac{\partial \ln c_{ns,m}}{\partial \ln k_1} & \frac{\partial \ln c_{ns,m}}{\partial \ln k_2} & \cdots & \frac{\partial \ln c_{ns,m}}{\partial \ln k_{np}} \end{pmatrix}. \tag{10}$$

$\widetilde{\boldsymbol{S}}$ is, thus, a matrix with a dimension of $(nt \cdot ns) \times np$. The eigenvalue-eigenvector decomposition technique is then applied on the matrix $\widetilde{\boldsymbol{S}}^{\mathrm{T}} \widetilde{\boldsymbol{S}}$ in Eq. (8), which leads to

$$\widetilde{\boldsymbol{S}}^{\mathrm{T}} \widetilde{\boldsymbol{S}} = \boldsymbol{U} \boldsymbol{\Lambda} \boldsymbol{U}^{\mathrm{T}}, \tag{11}$$

where $\boldsymbol{U}$ is a matrix of normed eigenvectors of $\widetilde{\boldsymbol{S}}^{\mathrm{T}} \widetilde{\boldsymbol{S}}$. $\boldsymbol{\Lambda}$ represents a diagonal matrix consisting of the eigenvalues $\lambda_1, \lambda_2, \ldots, \lambda_{np}$ of $\widetilde{\boldsymbol{S}}^{\mathrm{T}} \widetilde{\boldsymbol{S}}$. By replacing $\widetilde{\boldsymbol{S}}^{\mathrm{T}} \widetilde{\boldsymbol{S}}$ in Eq. (8) with the expression shown in Eq. (11), we obtain the approximate response function in the form of

$$\widetilde{Q}(\boldsymbol{\alpha}) = (\Delta\boldsymbol{\alpha})^{\mathrm{T}} \boldsymbol{U} \boldsymbol{\Lambda} \boldsymbol{U}^{\mathrm{T}} (\Delta\boldsymbol{\alpha}). \tag{12}$$

We then define the principal component $\boldsymbol{\Psi} = \boldsymbol{U}^{\mathrm{T}} \boldsymbol{\alpha}$ in order to write $\boldsymbol{U}^{\mathrm{T}} (\Delta\boldsymbol{\alpha})$ in the above equation as a variation of the
principal component,

$$\Delta\boldsymbol{\Psi} = \boldsymbol{U}^{\mathrm{T}} (\Delta\boldsymbol{\alpha}). \tag{13}$$



As a result of the definition of the principal component, Eq. (12) becomes

$$\widetilde{Q}(\boldsymbol{\alpha}) = \widetilde{Q}(\boldsymbol{\Psi}) = \sum_{j=1}^{np} \lambda_j \|\Delta\Psi_j\|^2, \tag{14}$$

wherein $\|\Delta\Psi_j\|^2 = (\Delta\Psi_j)^T(\Delta\Psi_j)$. From Eq. (14), it can be seen that the eigenvalue $\lambda_j$ depicts the significance of a group of reactions for the change of the system. $\Delta\Psi_j$ in Eq. (14) consists of $np$ elements which correspond to closely interacting reactions in the original complex reaction system. It denotes that the species concentrations are not only influenced by a separate individual reaction but a closely intertwining reaction sequence. Moreover, the contribution of the individual reaction to its group is also indicated by the weight of the corresponding element in the eigenvectors. Thus, by performing the principal component analysis on the matrix $\widetilde{\boldsymbol{S}}^T\widetilde{\boldsymbol{S}}$, important reactions can be indicated if they correspond to a large element of a principal component with a large eigenvalue.

Therefore, after obtaining the relative sensitivity coefficient $\widetilde{S}_{ij}$ of the reaction mechanism representing the chemistry of ODEs, principal component analysis is performed on the matrix $\widetilde{\boldsymbol{S}}^T\widetilde{\boldsymbol{S}}$. In that way $np$ eigenvalues and the associated eigenvectors are obtained. In order to derive a reaction mechanism with reduced size, we remove the reactions belonging to the principal components with small eigenvalues. It has been proved in previous studies (Vajda et al., 1985) that if the eigenvalue $\lambda_j$ is smaller than $ns \times nt \times 10^{-4}$, the variation in the concentration of each chemical species in the system is less than 10% at each instance in time. This selection criterion is also adopted in the present study. Aside from this, in the principal components corresponding to large eigenvalues, elements with values less than 0.2 are also considered as unimportant and, thus, can be removed from the original mechanism. It was estimated that these elements contribute less than 4% to the total variation of concentrations (Vajda et al., 1985).

It should also be noted that by removing the $j$-th reaction from the original chemical system, we actually define the reaction rate constant of the $j$-th reaction $k_j = 0$, for which Eq. (6) is invalid. To solve this problem the parameter

$$\widetilde{\alpha}_j = \frac{k_j}{k_j^0}, \quad j = 1, ..., np \tag{15}$$

is introduced (Vajda et al., 1985). It can be easily deduced that $\widetilde{\alpha}_j$ and the parameter $\alpha_j$ discussed above yield the same normalized sensitivity $\widetilde{S}_{ij}$. Therefore, the investigation of $\widetilde{\boldsymbol{S}}^T\widetilde{\boldsymbol{S}}$ for $\alpha_j$ described in the present manuscript is also valid for $\widetilde{\alpha}_j$ in the simplification process of the reaction mechanism.

Thus, at first, the original reaction mechanism described above is implemented in the box model KINAL (Turányi, 1990a) to capture the temporal behavior of ozone and principal bromine species. The boundary layer height used in the model is defined as 200 m. This is considered to be a representative value since the boundary layer height under typical polar conditions ranges from 100 m to 500 m (Stull, 1988). The initial atmospheric composition used in the model is taken from the previous box model study (Cao et al., 2014), which is listed in Tab. 1. In the present model, the prescribed 0.3 ppt $Br_2$ and 0.01 ppt HBr play the role of triggering the bromine explosion mechanism and the associated ozone consumption. Emissions of nitrogen oxides ($NO_x$), HONO, $H_2O_2$ and HCHO from the underlying surface are also implemented in the model according to the measurements (Jones et al., 2000, 2001; Jacobi et al., 2002; Grannas et al., 2007), and listed in Tab. 2. The ratio of HONO and $NO_2$ is assumed to be unity (Grannas et al., 2007).





The photolysis reaction rates are estimated by using a three-stream radiation transfer model (Röth, 1992, 2002) with an assumption of SZA = 80° and a surface albedo unity. At present, the daily variation in SZA and the dark reactions are not accounted for in the model. The influences brought about by the inclusion of the change in SZA and the dark reactions have been proved to be small in previous investigations (Lehrer et al., 2004; Cao et al., 2014). The heterogeneous reactions representing the bromine recycling processes on the surfaces of the ice/snowpack and the suspended aerosols are also included in the original reaction mechanism of ODEs. It has been proved that the rates of these heterogeneous reactions critically control the bromine amount in the boundary layer and the depletion rate of ozone during ODEs (Cao et al., 2014). In the present study, the parameterizations of these heterogeneous reactions are also adopted from the box model study by Cao et al. (2014). Recently, a snowpack module representing the mass exchange between the snowpack and the ambient air has been developed by the author and co-workers (Cao et al., 2016b). However, this snowpack module was not applied in the reaction mechanism used in the present study.

Later, the concentration sensitivity analysis is applied on the original reaction mechanism of ODEs to identify the most influential reactions during different time periods. The temporal behavior of the concentration sensitivity coefficient for each reaction is also captured. Then the reactions with a maximum absolute value of the relative concentration sensitivity less than 10% are removed from the original reaction scheme so that a reduced reaction mechanism is obtained.

After the computation of the relative concentration sensitivity coefficients, the principal component analysis is performed on the matrix $\widetilde{\boldsymbol{S}}^{\mathrm{T}}\widetilde{\boldsymbol{S}}$. Principal components corresponding to small eigenvalues are regarded as making negligible contributions to the overall response of the whole system and, thus, can be eliminated. Since 38 chemical species (all species in the mechanism excluding $N_2$) and 94 time steps are focused on in the present simulation, the criterion of eigenvalues used for dividing important and unimportant principal components is calculated as $38 \times 94 \times 10^{-4} = 0.3572$. Moreover, as discussed above, if an element in a relatively important principal component has a value of less than 0.2, this element can be also removed due to its relatively small contribution to the total variation of the system.

In the following section, the most important computational results are presented and discussed.

## 3 Results and Discussion

At the beginning of this study, we ran the box model KINAL with the implementation of the original complex reaction scheme to capture the temporal change of principal chemical species within the 200 m boundary layer. Figure 1 displays the development of the mixing ratios of ozone and principal bromine species with time. According to previous studies (Cao et al., 2014), the whole ODE can be divided into three periods. In the first period named "induction stage" which corresponds to the time period before day 3 in Fig. 1, the consumption rate of ozone in the boundary layer is less than 0.1 ppb h$^{-1}$. Moreover, the mixing ratio of ozone remains steady at a background level ($\sim$ 40 ppb). Due to the presence of ozone in this time period, the formed Br atoms are rapidly oxidized to be BrO, and thus can be hardly observed at this time stage. In contrast to that, the mixing ratios of HOBr and BrO steadily increase within this time period, which makes them the major bromine containing compounds.



After day 3 (see Fig. 1), as bromide is continuously activated from the ice/snow surface via the bromine explosion mechanism, the total bromine loading in the ambient air keeps increasing. As a result, the depletion rate of ozone exceeds 0.1 ppb h$^{-1}$, and the mixing ratio of ozone declines rapidly until a value lower than 10% of its original level (4 ppb) is achieved, on approximately day 4.6. This period (from day 3 to day 4.6) is thus named "depletion stage". Within the depletion stage, the mixing

ratios of BrO and HOBr reach their peaks and then drop instantaneously to a level less than 1 ppt due to the nearly complete removal of ozone in the boundary layer.

After the depletion stage, the last time period of the ODE is the "end stage" in which the ozone mixing ratio is lower than 4 ppb. At the beginning of the end stage, Br atom becomes the most dominant brominated species and its mixing ratio increases to a peak value of approximately 170 ppt (see the black solid line in Fig. 1). This large amount of Br is then absorbed by the

aldehydes in the atmosphere, and the total bromine amount left in the ambient air becomes steady towards the end of the simulation (see the purple solid line in Fig. 1). After day 6, the dominant bromine species in the ambient air is mostly in the form of HBr, which is in consistence with the previous finding (Langendörfer et al., 1999). The features of the temporal change of these chemical species are similar to those obtained in previous studies (Lehrer et al., 2004; Cao et al., 2014, 2016a, b) except a slight difference in the onsets of the aforementioned time periods. The discrepancy between the results from different model

studies is attributed to the extension of the previous reaction mechanism in the present study.

After capturing the temporal behavior of principal chemical species, the original reaction mechanism consisting of 39 species and 92 reactions is processed with the concentration sensitivity analysis, which is shown in the next subsection.

### 3.1    Concentration Sensitivity Analysis of the Reaction Mechanism of the ODE

The relative concentration sensitivity coefficient of each chemical species for all the reactions in the original mechanism within

each time step is calculated by performing the concentration sensitivity analysis. Figure 2 depicts the sensitivity coefficients of ozone and BrO for all the reactions in the mechanism within the time interval [day 3.9, day 4] which resides within the depletion stage. It can be seen that the sensitivity of the mixing ratios of ozone and BrO to each chemical reaction in the mechanism is clearly indicated by the values of the sensitivity coefficients.

In Fig. 2(a), reaction (R15), which denotes the heterogeneous bromine activation from the ice/snow-covered surfaces, is

identified as the most important reaction for both ozone and BrO mixing ratios at this time stage. This is related to the fact that during the depletion stage, the HOBr concentration increases vigorously until its peak value is achieved. Thus, a large amount of bromide is activated from the ice/snow surfaces. As a result, reaction (R15) critically determines the total bromine amount in the air and the depletion rate of ozone in this time period. Due to the same reason, reactions with the involvement of HOBr, (R10) and (R11), play significant roles in influencing the mixing ratios of ozone and BrO, which is indicated in Fig. 2(a).

Aside from the reactions associated with HOBr, reactions which are able to produce or consume Br atoms are also influential since Br reacts with ozone directly and rapidly. Thus, the significance of the Br associated reactions, (R5), (R7), (R17) and (R20), is displayed in Fig. 2(a) according to the relatively larger absolute values of the sensitivity coefficients. By evaluating the sensitivities at an earlier time instance (not shown here), we found that the Br associated reactions are the most dominant





reactions for ozone and BrO in the induction stage. However, from Fig. 2(a), it can be concluded that during the depletion stage, the importance of the HOBr associated reactions exceed the ones with the involvement of Br atoms.

In Fig. 2(b), it is shown that the nitrogen related reactions are less important for the mixing ratios of ozone and BrO in the boundary layer compared to the bromine related reactions. However, the moderate role of the hydrolysis reaction of BrONO$_2$, (R90), is depicted. The reason is attributable to the formation of HOBr in this reaction, which may also speed up the bromine activation during this time period. Moreover, we found that the reactions with the involvement of BrONO$_2$ are of more importance during the end stage, which is consistent with the previous box model study (Cao et al., 2014).

The results discussed above show that the values of the concentration sensitivity coefficients vary with time. This reveals that the importance of each reaction differs significantly during different time periods. In order to clarify the temporal behavior of the concentration sensitivities, we also captured the development of the sensitivities of ozone for the dominant reactions with time (see Fig. 3). As similar as the results shown in Fig. 2, it is seen in Fig. 3 that the HOBr associated reactions, (R10), (R11) and (R15), are the most influential reactions for the ozone mixing ratio during the depletion stage and the beginning of the end stage as well. The maximum absolute values of the sensitivities corresponding to reactions (R10), (R11) and (R15) are 71, 78 and 164, respectively. The second most important reactions are the Br associated reactions, (R5), (R7), (R17) and (R20) (see Fig. 3). However, the values of their sensitivity coefficients are much lower than those of the HOBr related reactions during the time period under investigation.

Aiming to reduce the size of the original reaction mechanism, we summarize the maximum absolute values of the concentration sensitivities for each reaction in the original mechanism which are listed in Tab. 3. Then the reactions with the maximum absolute value lower than 10% are removed from the mechanism. As a result, 11 reactions, namely (R18), (R30), (R49), (R52), (R56), (R67), (R71), (R73), (R74), (R79) and (R80), are eliminated from the original reaction mechanism of ODEs. The reduced reaction mechanism obtained after the simplification process, thus, contains 39 species and 81 reactions in total.

We applied the reduced reaction mechanism in the box model KINAL to see the deviations from the original reaction scheme. Figure 4 shows a comparison of the mixing ratios of ozone and principal bromine species computed by using the original reaction mechanism and the reduced reaction mechanism. It is found that both simulations give nearly identical results. The depletion stage of the ODE predicted by the reduced reaction mechanism lasts from day 3 to day 4.6 which is consistent with the results obtained by using the original reaction scheme. The maximum deviations of the mixing ratios of Br, HBr, HOBr, BrO and Br$_{tot}$ between the original reaction mechanism and the reduced reaction mechanism are 0.6%, 0.5%, 0.6%, 0.1% and 0.2%, respectively. Thus, it can be seen that the calculated concentration changes of ozone and principal bromine species from these two mechanisms agree very well, within deviations of 1%.

The principal component analysis was then performed on the sensitivity matrix $\widetilde{S}^{\mathrm{T}}\widetilde{S}$ of the original reaction scheme to simplify the mechanism, which is shown in the next subsection.





## 3.2 Principal Component Analysis of the Reaction Mechanism of the ODE

After obtaining the concentration sensitivities of the original reaction mechanism of the ODE, the matrix $\widetilde{S}^T \widetilde{S}$ is constructed and applied in the principal component analysis. The eigenvalues and the corresponding principal components of $\widetilde{S}^T \widetilde{S}$ are thus obtained which are listed in Tab. 4. The selection criterion $\lambda_i < 0.3572$ is adopted for removing reactions from the scheme

which has been discussed above. Moreover, in the principal components with large eigenvalues, if an element contributes less than 0.2 to the corresponding principal component, the reaction associated with this element can be also removed. Thus, by summarizing all the reactions which are identified as redundant according to the principal component analysis, 20 reactions, (R2), (R18), (R27), (R28), (R30), (R38), (R40), (R43), (R49), (R52), (R56), (R67), (R71), (R72), (R73), (R74), (R77), (R79), (R80) and (R87), can be removed from the original reaction scheme, which are indicated in Tab. 3 by the reaction numbers in

boxes.

It is found that the reactions identified as unimportant by using the sensitivity analysis are all covered by the principal component analysis. Moreover, 9 extra reactions to be eliminated from the reaction scheme are also revealed by the principal component analysis. The reason for the difference between these two approaches is possibly that in a complex reaction mechanism, usually the removal of one single reaction would cause significant variations in the temporal change of many chemical

species while the elimination of a group of reactions may have only minor impact on the response of the system. In the concentration sensitivity analysis, the association between a particular species concentration and each reaction rate is clarified, which is suitable for screening reactions separately. In contrast to that, the principal component analysis is able to identify the dependence of the species concentrations on a group of reactions. Therefore, by performing the principal component analysis, we were able to remove more reactions from the original reaction mechanism. At last, the reduced reaction mechanism after

the implementation of the principal component analysis consists of 39 species and 72 reactions (see Tab. 3).

After performing the principal component analysis, the reduced reaction mechanism was introduced into the box model to simulate the mixing ratio change of ozone and principal bromine species (see Fig. 5). It is seen that the results obtained by using the reduced reaction mechanism agree well with those simulated by using the original reaction mechanism. The depletion stage in simulation results using the reduced reaction mechanism starts on day 2.9 and finishes on day 4.4, which occurs a little bit

earlier than that using the original reaction mechanism. The maximum deviations of Br, HBr, HOBr, BrO and $Br_{tot}$ are 6.6%, 1.6%, 8.9%, 2.5% and 5.2%, respectively. Thus, the variations in the mixing ratios of the principal bromine species considered in the model caused by the removal of 20 redundant reactions are less than 10% which is restrained by the selection criterion adopted. Therefore, the reduced reaction mechanism can satisfactorily capture the temporal change of each chemical species so that the requirement of the accuracy of the reaction mechanism is fulfilled for multi-dimensional computations of ODEs.

The principal component analysis was applied to different time periods of ODEs (induction stage, depletion stage and end stage), and the computational results are displayed in Fig. 6. The red contours denote the important reactions in this time period while the blue contours show the reactions which can be removed. Similarly, we summarize all the important reactions at different time stages and remove the least significant reactions. It is found that 20 reactions, (R2), (R18), (R27), (R28), (R29), (R30), (R38), (R40), (R43), (R45), (R49), (R52), (R56), (R67), (R71), (R73), (R74), (R77), (R79) and (R80), can be removed





according to the principal component analysis for different time periods. It should be noticed that reactions (R29) and (R45) which are identified as important in the global principal component analysis are currently indicated as redundant. It is because of that although reactions (R29) and (R45) play minor roles and the removal of them causes less than 10% of the variation in the system response within each time period, from a global view of the complete ODE, the variations within each time period

would accumulate, which leads to a total concentration change exceeding 10% of the original value. Therefore, reactions (R29) and (R45) cannot be removed if the selection criterion is strictly obeyed in the global principal component analysis.

Moreover, it is also found that reactions (R72) and (R87) which used to be identified as unimportant are now considered as important, and cannot be removed from the reaction scheme. From the computational results of the principal component analysis for different time periods (see Fig. 6b), it is observed that reactions (R72) and (R87) are all identified as important in

the induction stage while their significance during the depletion stage and the end stage are negligible. Thus, in the principal component analysis when the whole depletion event is observed, the importance of (R72) and (R87) are smoothed out. As a result, these two reactions can be removed from the original reaction scheme.

Apart from screening reactions from a complex reaction mechanism, principal component analysis is also capable to clarify the chemical species in quasi-steady state which are indicated by principal components with small eigenvalues. In order to

perform this investigation, the principal component analysis was once more applied on the reduced reaction mechanism with 39 species and 72 reactions. By observing the principal components with small eigenvalues (not shown here), we found that No. 71 principal component which corresponds to the second least eigenvalue has only two dominant elements, (R1): $O_3 + h\nu \rightarrow O(^1D) + O_2$ and (R3): $O(^1D) + N_2 \xrightarrow{O_2} O_3 + N_2$. Besides, the value of the element (R1), 0.710, is approximately equal to that of (R3) which is 0.704. We also noticed that (R1) and (R3) are the reactions in which the chemical species $O^1(D)$ participate.

Thus, it represents that $O^1(D)$ is in quasi-steady state, and the computational results depend only on the ratio of the reaction rates of (R1) and (R3), which can be expressed as

$$k_s = \frac{k_1}{k_3}.$$

(16)

Herein, $k_1$ and $k_3$ are the rate constants of reactions (R1) and (R3), respectively. In order to confirm this finding, we multiplied the rate coefficient of both (R1) and (R3) by 100, the deviations of the system after this change are shown in Tab. 5. It is found

that the peak values of the principal bromine species calculated from the reduced reaction mechanism and from the reaction scheme with the modified reaction rates agree well within 1 percent. In contrast to that, if we only increase the rate constant of (R1) to 100 times of its original value while the rate constant of (R3) remains the same, the difference between the results from the original mechanism and the modified mechanism becomes larger (see Tab. 5). The maximum deviation amounts to 172% for the species HOBr. This finding also confirms the validity of the principal component analysis.

As $O^1(D)$ was found in quasi-steady state during ODEs, it is possible to directly estimate the mixing ratio of $O^1(D)$ at each time instance according to the mixing ratios of other chemical species. It can be easily deduced that the mixing ratio of $O^1(D)$ during the depletion event is expressed as

$$[O^1(D)] = \frac{k_1[O_3]}{k_3[N_2]},$$

(17)



in which $[O_3]$ and $[N_2]$ are the mixing ratios of ozone and nitrogen, respectively. Thus, in multi-dimensional model studies of ODEs, aside from using a reduced reaction mechanism to shorten the time cost for the calculation of the chemical source terms, the species equation corresponding to $O^1(D)$ can be also neglected, which improves the efficiency of the computations. However, the simplification of the reaction mechanism by using the quasi-steady state approximation is beyond the scope of the present study.

## 4   Conclusions and Future Developments

In the present study, two reduction approaches, namely the concentration sensitivity analysis and the principal component analysis, were applied on a reaction mechanism representing the chemistry of ODEs. The former was performed based on the ratio of the relative change of species concentrations and the variation of a particular reaction rate. The significance of each reaction in the original mechanism for various chemical species was identified. It was found that during the depletion of ozone, reactions associated with HOBr are the most influential reactions as they determine the total bromine loading in the atmosphere within this time period. By removing 11 reactions which exhibit peak absolute values of the sensitivity coefficients lower than 10%, a reduced reaction mechanism of ODEs was derived. The difference between the results obtained by using the original reaction mechanism and the reduced reaction mechanism is negligible, which validates the sensitivity analysis.

The principal component analysis, on the other hand, is able to extract inherent information from a chemical kinetic system via the eigenvalue decomposition of the matrix $\widetilde{S}^{\mathrm{T}} \widetilde{S}$. In the herein presented study, the reactions which have principal components with eigenvalues lower than a critical criterion were eliminated from the original reaction mechanism. Additionally, the elements which occupy less than 0.2 of each principal component were also removed. In total, 20 reactions were identified redundant using the principal component analysis and, thus, screened out from the original ODE mechanism. As a result, a reaction scheme with a reduced size consisting of 39 species and 72 reactions was obtained. The deviations of the mixing ratios of ozone and principal bromine species between the original reaction mechanism and the reduced reaction mechanism after the principal component analysis are within 10%. This proves the suitability of the obtained reduced reaction mechanism for multi-dimensional simulations of ODEs. Apart from this, displayed by the principal components with the smallest eigenvalues, the chemical species $O^1(D)$ is identified to be in quasi-steady state. This facilitates a further improvement of the numerical efficiency in high-dimensional simulations of ODEs.

It is obvious that the applications of the concentration sensitivity analysis and the principal component analysis are not restricted to the reaction mechanism of ODEs which is the scope of the present study. Any chemical kinetic system which is expressible through a form of Eq. (1) can be analyzed and optimized by using these two approaches. Furthermore, the readers might have noticed that the two reduction approaches used in the present study do not lead to the removal of redundant species from the detailed reaction mechanism. The reason is that they, rather than employing erroneous simplifications (Vajda et al., 1985; Turányi, 1990b), focus on all occurring chemical species. This, however, prohibits the identification of redundant species. Therefore, it seems worth to utilize another approach in order to further reduce the reaction mechanism of the ODE after the analyses performed herein. At present, three types of reduction methods are normally used in previous studies (Turányi





and Tomlin, 2014): species lumping, timescale-separation method, and tabulation approaches. As discussed above, the quasi-steady state approximation, which belongs to the category of the timescale-separation method, is capable of determining the concentrations of the species in quasi-steady state through a function of other species concentrations, which helps to remove the species in quasi-steady state from the kinetic reaction scheme. Apart from this, currently, another type of a timescale-based

5  analysis approach, the so-called intrinsic low-dimensional manifolds (ILDM) (Maas and Pope, 1992), is being implemented by the authors of this paper. By means of the ILDM, the dimension of the composition space can be reduced significantly based on the number of the degrees of freedom. As a result, the computational effort of the calculation of the chemical source terms in the rate equations is decreased remarkably, what may further progress the current work.

*Acknowledgements.* The authors sincerely thank the financial supports provided by the National Natural Science Foundation of China

10  (No. 41375044), the Natural Science Foundation of Jiangsu Province (No. 2015s042), the Double Innovation Talent Program (No. R2015SCB02), the Polar Strategic Foundation (No. 20150308) and the Startup Foundation for Introducing Talent of NUIST (No. 2014r066).



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



**Table 1.** Initial atmospheric composition in the boundary layer (ppm = parts per million, ppb = parts per billion, ppt = parts per trillion) (Cao et al., 2014).

| Species | Mixing ratio |
|---------|-------------|
| $O_3$ | 40 ppb |
| $Br_2$ | 0.3 ppt |
| HBr | 0.01 ppt |
| $CH_4$ | 1.9 ppm |
| $CO_2$ | 371 ppm |
| CO | 132 ppb |
| HCHO | 100 ppt |
| $CH_3CHO$ | 100 ppt |
| $C_2H_6$ | 2.5 ppb |
| $C_2H_4$ | 100 ppt |
| $C_2H_2$ | 600 ppt |
| $C_3H_8$ | 1.2 ppb |
| NO | 5 ppt |
| $NO_2$ | 10 ppt |
| $H_2O$ | 800 ppm |

**Table 2.** Emission fluxes from the underlying surface (Cao et al., 2014).

| Species | Emission rates [molec. $cm^{-2} s^{-1}$] | Reference |
|---------|-------------------------------------------|-----------|
| NO | $1.6 \times 10^7$ | Jones et al. (2000, 2001) |
| $NO_2$ | $1.6 \times 10^7$ | Jones et al. (2000, 2001) |
| HONO | $1.6 \times 10^7$ | Grannas et al. (2007) |
| $H_2O_2$ | $1.0 \times 10^8$ | Jacobi et al. (2002) |
| HCHO | $6.0 \times 10^7$ | Jacobi et al. (2002) |





**Table 3.** The maximum absolute value of the concentration sensitivity coefficient for each reaction in the mechanism and the time when the maximum value occurs. Reaction number with double underlines denotes that this reaction is identified as unimportant via the concentration sensitivity analysis, and reaction number in a box represents that this reaction is indicated as unimportant via the principal component analysis.

| Reaction Number | Sensitivity Maximum | Occurring Day | Reaction Number | Sensitivity Maximum | Occurring Day | Reaction Number | Sensitivity Maximum | Occurring Day |
|---|---|---|---|---|---|---|---|---|
| (R1) | $3.1 \times 10^1$ | 4.75 | (R32) | $2.0 \times 10^1$ | 4.75 | (R63) | $1.5 \times 10^1$ | 4.75 |
| (R2) | $8.9 \times 10^0$ | 4.75 | (R33) | $9.6 \times 10^0$ | 4.75 | (R64) | $6.9 \times 10^0$ | 4.75 |
| (R3) | $2.2 \times 10^1$ | 4.75 | (R34) | $1.0 \times 10^0$ | 5.0 | (R65) | $7.9 \times 10^{-1}$ | 0.1 |
| (R4) | $3.1 \times 10^1$ | 4.75 | (R35) | $1.0 \times 10^0$ | 4.6 | (R66) | $1.5 \times 10^0$ | 4.75 |
| (R5) | $1.1 \times 10^2$ | 4.75 | (R36) | $1.2 \times 10^0$ | 4.8 | (R67) | $1.3 \times 10^{-2}$ | 0.1 |
| (R6) | $6.1 \times 10^0$ | 4.75 | (R37) | $2.3 \times 10^0$ | 4.75 | (R68) | $4.7 \times 10^{-1}$ | 4.3 |
| (R7) | $8.7 \times 10^1$ | 4.75 | (R38) | $4.1 \times 10^0$ | 4.75 | (R69) | $1.8 \times 10^0$ | 2.1 |
| (R8) | $2.0 \times 10^1$ | 4.75 | (R39) | $2.5 \times 10^1$ | 4.75 | (R70) | $2.9 \times 10^0$ | 2.1 |
| (R9) | $3.2 \times 10^0$ | 10.0 | (R40) | $2.7 \times 10^0$ | 4.75 | (R71) | $9.2 \times 10^{-2}$ | 4.75 |
| (R10) | $1.3 \times 10^2$ | 4.75 | (R41) | $4.8 \times 10^0$ | 2.0 | (R72) | $4.0 \times 10^{-1}$ | 4.75 |
| (R11) | $1.4 \times 10^2$ | 4.75 | (R42) | $3.8 \times 10^1$ | 4.75 | (R73) | $8.6 \times 10^{-5}$ | 4.75 |
| (R12) | $3.9 \times 10^1$ | 4.75 | (R43) | $4.0 \times 10^0$ | 4.75 | (R74) | $5.2 \times 10^{-2}$ | 4.75 |
| (R13) | $9.9 \times 10^0$ | 10.0 | (R44) | $1.8 \times 10^0$ | 10.0 | (R75) | $1.7 \times 10^1$ | 4.75 |
| (R14) | $6.0 \times 10^1$ | 4.75 | (R45) | $1.0 \times 10^0$ | 10.0 | (R76) | $9.0 \times 10^{-1}$ | 2.1 |
| (R15) | $2.9 \times 10^2$ | 4.75 | (R46) | $3.8 \times 10^0$ | 10.0 | (R77) | $1.1 \times 10^{-1}$ | 6.6 |
| (R16) | $1.7 \times 10^1$ | 4.75 | (R47) | $9.8 \times 10^{-1}$ | 0.1 | (R78) | $1.9 \times 10^1$ | 4.75 |
| (R17) | $8.2 \times 10^1$ | 4.75 | (R48) | $7.9 \times 10^{-1}$ | 10.0 | (R79) | $3.9 \times 10^{-8}$ | 10.0 |
| (R18) | $8.7 \times 10^{-2}$ | 10.0 | (R49) | $3.5 \times 10^{-2}$ | 4.75 | (R80) | $6.0 \times 10^{-5}$ | 4.75 |
| (R19) | $2.3 \times 10^0$ | 4.75 | (R50) | $1.6 \times 10^0$ | 7.2 | (R81) | $1.4 \times 10^0$ | 4.75 |
| (R20) | $1.0 \times 10^2$ | 4.75 | (R51) | $1.0 \times 10^0$ | 4.8 | (R82) | $1.0 \times 10^1$ | 4.75 |
| (R21) | $2.5 \times 10^1$ | 10.0 | (R52) | $4.3 \times 10^{-2}$ | 7.0 | (R83) | $1.0 \times 10^1$ | 4.75 |
| (R22) | $5.7 \times 10^1$ | 4.75 | (R53) | $2.4 \times 10^0$ | 4.75 | (R84) | $2.6 \times 10^0$ | 4.75 |
| (R23) | $1.2 \times 10^1$ | 4.8 | (R54) | $2.0 \times 10^0$ | 4.75 | (R85) | $3.3 \times 10^{-1}$ | 4.8 |
| (R24) | $2.2 \times 10^0$ | 2.0 | (R55) | $1.3 \times 10^1$ | 8.7 | (R86) | $2.7 \times 10^0$ | 2.0 |
| (R25) | $3.3 \times 10^0$ | 4.75 | (R56) | $3.4 \times 10^{-4}$ | 4.75 | (R87) | $8.0 \times 10^0$ | 4.75 |
| (R26) | $2.0 \times 10^1$ | 4.75 | (R57) | $1.7 \times 10^1$ | 4.75 | (R88) | $2.3 \times 10^0$ | 4.75 |
| (R27) | $2.5 \times 10^0$ | 4.75 | (R58) | $7.1 \times 10^1$ | 4.75 | (R89) | $3.6 \times 10^0$ | 7.0 |
| (R28) | $3.9 \times 10^0$ | 4.75 | (R59) | $3.7 \times 10^1$ | 4.75 | (R90) | $1.4 \times 10^1$ | 4.75 |
| (R29) | $2.1 \times 10^{-1}$ | 4.75 | (R60) | $2.3 \times 10^1$ | 4.75 | (R91) | $1.4 \times 10^0$ | 10.0 |
| (R30) | $1.4 \times 10^{-4}$ | 4.75 | (R61) | $1.1 \times 10^1$ | 4.75 | (R92) | $1.6 \times 10^0$ | 4.75 |
| (R31) | $2.9 \times 10^1$ | 4.75 | (R62) | $2.0 \times 10^0$ | 4.75 | | | |





Atmospheric Chemistry and Physics Discussions — Open Access

**Table 4.** Eigenvalues ($\lambda$) and the correponding principal components of $\widetilde{S}^T \widetilde{S}$ for the original reaction mechanism of the ODE. Top line in the column of $\lambda$ refers to the number of the eigenvalue, and the bottom line gives the value of $\lambda$. Here only the principal components with eigenvalues larger than 0.3572 are listed. Within each principal component, elements larger than 0.2 are displayed.

| # | $\lambda$ | Dominant elements of the principal components |
|---|-----------|-----------------------------------------------|
| 1 | $1.9 \times 10^6$ | (R15) (R11) (R20) (R10) (R5) |
| 2 | $2.3 \times 10^5$ | (R20) (R5) (R21) (R7) (R11) (R8) |
| 3 | $3.4 \times 10^3$ | (R20) (R83) (R39) (R15) (R12) (R75) (R11) |
| 4 | $2.6 \times 10^3$ | (R83) (R20) (R11) (R75) (R24) (R82) (R8) (R41) |
| 5 | $2.1 \times 10^3$ | (R5) (R21) (R7) (R15) (R39) (R20) (R13) |
| 6 | $1.3 \times 10^3$ | (R41) (R12) (R21) (R31) (R83) (R86) (R91) |
| 7 | $5.9 \times 10^2$ | (R21) (R5) (R24) (R12) (R7) (R31) |
| 8 | $4.6 \times 10^2$ | (R12) (R33) (R10) (R35) (R41) (R19) |
| 9 | $3.4 \times 10^2$ | (R70) (R55) (R19) (R69) (R62) (R13) (R33) (R83) (R35) |
| 10 | $3.0 \times 10^2$ | (R35) (R33) (R10) (R24) (R17) (R15) (R86) (R83) (R8) |
| 11 | $2.9 \times 10^2$ | (R24) (R89) (R35) (R70) (R86) (R19) (R55) (R47) (R84) (R33) |
| 12 | $2.7 \times 10^2$ | (R89) (R84) (R70) (R75) (R10) (R23) |
| 13 | $2.4 \times 10^2$ | (R86) (R21) (R89) (R24) (R20) (R90) |
| 14 | $2.0 \times 10^2$ | (R70) (R75) (R1) (R69) (R51) (R24) (R3) (R53) |
| 15 | $1.6 \times 10^2$ | (R14) (R1) (R51) (R3) (R35) (R55) (R15) (R62) (R39) (R12) |
| 16 | $1.4 \times 10^2$ | (R23) (R57) (R22) (R75) (R6) (R53) (R86) (R42) |
| 17 | $1.2 \times 10^2$ | (R53) (R10) (R51) (R17) (R19) (R76) (R88) (R90) |
| 18 | $1.1 \times 10^2$ | (R6) (R1) (R53) (R32) (R88) (R3) (R55) (R13) (R76) |
| 19 | $9.7 \times 10^1$ | (R42) (R88) (R76) (R17) (R32) |
| 20 | $8.8 \times 10^1$ | (R76) (R88) (R51) (R42) (R14) (R57) (R17) |
| 21 | $8.3 \times 10^1$ | (R6) (R14) (R32) (R19) (R55) (R51) (R23) (R47) (R75) |
| 22 | $7.5 \times 10^1$ | (R17) (R42) (R88) (R11) (R76) (R57) (R55) (R6) |
| 23 | $6.9 \times 10^1$ | (R6) (R1) (R14) (R75) (R26) (R3) |
| 24 | $5.9 \times 10^1$ | (R31) (R10) (R39) (R23) (R11) |
| 25 | $5.5 \times 10^1$ | (R53) (R62) (R51) (R55) (R26) (R39) (R17) |
| 26 | $4.9 \times 10^1$ | (R23) (R58) (R59) (R13) (R90) (R62) (R46) |
| 27 | $4.0 \times 10^1$ | (R90) (R57) (R13) (R55) (R47) (R53) (R24) (R14) |
| 28 | $3.7 \times 10^1$ | (R34) (R46) (R26) (R8) (R90) (R13) (R62) |
| 29 | $3.4 \times 10^1$ | (R53) (R47) (R58) (R34) (R26) (R12) (R24) |
| 30 | $3.0 \times 10^1$ | (R90) (R34) (R57) (R26) |
| 31 | $2.8 \times 10^1$ | (R34) (R11) (R13) (R14) (R78) (R26) |
| 32 | $2.5 \times 10^1$ | (R13) (R58) (R50) (R55) (R59) (R31) |





| $\lambda$ | Dominant elements of the principal components | | | | |
|---|---|---|---|---|---|
| 33 $2.4 \times 10^1$ | (R59) (R23) | (R34) | (R58) | (R26) | (R78) |
| 34 $2.2 \times 10^1$ | (R46) | (R59) | (R34) | (R58) | |
| 35 $1.9 \times 10^1$ | (R8) | (R62) | (R51) | (R57) | |
| 36 $1.7 \times 10^1$ | (R47) (R8) | (R19) (R81) | (R62) | (R31) | (R51) |
| 37 $1.5 \times 10^1$ | (R16) (R31) | (R63) (R12) | (R39) | (R26) | (R82) |
| 38 $1.3 \times 10^1$ | (R8) (R12) | (R32) (R59) | (R57) | (R50) | (R42) |
| 39 $1.2 \times 10^1$ | (R84) (R83) | (R89) (R75) | (R91) (R81) | (R8) | (R86) |
| 40 $1.1 \times 10^1$ | (R16) | (R19) | (R81) | (R63) | (R84) |
| 41 $9.6 \times 10^0$ | (R91) | (R84) | (R89) | | |
| 42 $8.8 \times 10^0$ | (R63) | (R16) | (R41) | (R65) | (R59) |
| 43 $8.1 \times 10^0$ | (R50) | (R13) | (R9) | (R55) | (R47) |
| 44 $7.4 \times 10^0$ | (R68) | (R81) | (R63) | (R60) | (R78) |
| 45 $7.3 \times 10^0$ | (R84) (R61) | (R9) (R78) | (R81) (R82) | (R41) | (R60) |
| 46 $5.6 \times 10^0$ | (R65) | (R68) | (R88) | (R78) | (R76) |
| 47 $5.0 \times 10^0$ | (R84) (R70) | (R75) (R86) | (R31) | (R83) | (R32) |
| 48 $4.9 \times 10^0$ | (R68) | (R9) | (R65) | (R81) | |

| $\lambda$ | Dominant elements of the principal components | | | | |
|---|---|---|---|---|---|
| 49 $4.3 \times 10^0$ | (R9) (R78) | (R65) (R91) | (R61) | (R26) | (R63) |
| 50 $3.6 \times 10^0$ | (R60) (R82) | (R31) (R75) | (R84) (R86) | (R83) | (R32) |
| 51 $3.0 \times 10^0$ | (R7) (R81) | (R5) (R36) | (R61) | (R22) | (R31) |
| 52 $2.4 \times 10^0$ | (R69) | (R70) | (R7) | (R60) | (R5) |
| 53 $1.9 \times 10^0$ | (R22) | (R25) | (R5) | (R7) | (R36) |
| 54 $1.8 \times 10^0$ | (R81) (R70) | (R69) (R63) | (R82) (R5) | (R7) | (R61) |
| 55 $1.6 \times 10^0$ | (R36) | (R25) | (R37) | (R69) | |
| 56 $1.5 \times 10^0$ | (R48) (R36) | (R64) (R25) | (R22) | (R61) | (R78) |
| 57 $1.0 \times 10^0$ | (R48) | (R25) | (R22) | (R61) | |
| 58 $9.7 \times 10^{-1}$ | (R25) | (R48) | (R64) | (R54) | (R44) |
| 59 $7.2 \times 10^{-1}$ | (R54) (R81) | (R44) (R22) | (R25) | (R64) | (R66) |
| 60 $6.4 \times 10^{-1}$ | (R66) (R82) | (R64) (R29) | (R4) | (R61) | (R78) |
| 61 $4.7 \times 10^{-1}$ | (R85) | (R66) | | | |
| 62 $4.3 \times 10^{-1}$ | (R66) (R88) | (R4) | (R85) | (R76) | (R65) |
| 63 $3.7 \times 10^{-1}$ | (R54) | (R44) | (R45) | (R48) | |





**Table 5.** Comparison of peak values of principal bromine species calculated by using various chemical reaction mechanisms. The peak values shown in the table are calculated by using the reduced reaction mechanism derived after the principal component analysis.

| Species | Peak Value [ppt] | $k_1 \times 100, k_3 \times 100$ Deviation [%] | $k_1 \times 100, k_3 \times 1$ Deviation [%] |
|---|---|---|---|
| Br | 177.6 | 0.3 | 104.4 |
| HBr | 203.7 | 0.1 | 126.8 |
| HOBr | 95.6 | 0.7 | 172.0 |
| BrO | 58.1 | 0.1 | 63.7 |
| $Br_{tot}$ | 203.8 | 0.1 | 126.7 |





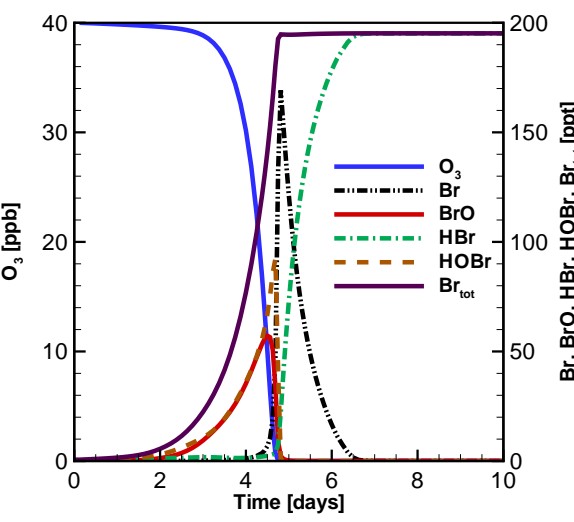

**Figure 1.** Temporal evolution of ozone and principal bromine containing compounds in a 200 m boundary layer, obtained by using the original reaction scheme in the model.





(a)

**Figure 2.** Relative concentration sensitivities of ozone and BrO for (a) reactions (R1)-(R46) and (b) reactions (R47)-(R92) within the time interval [day 3.9, day 4]. (Continued...)





(b)

**Figure 2.** Relative concentration sensitivities of ozone and BrO for (a) reactions (R1)-(R46) and (b) reactions (R47)-(R92) within the time interval [day 3.9, day 4].





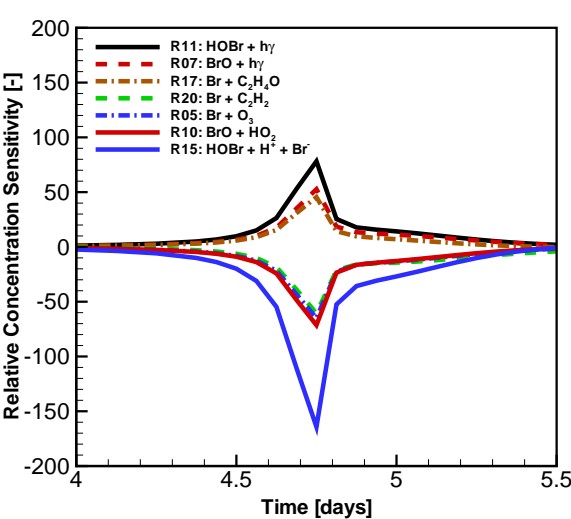

**Figure 3.** Temporal change of the relative concentration sensitivities of ozone for the dominant reactions between day 4 and day 5.5. Here only the reactions with a maximum absolute value of the relative concentration sensitivities larger than 40 are identified as dominant and shown.





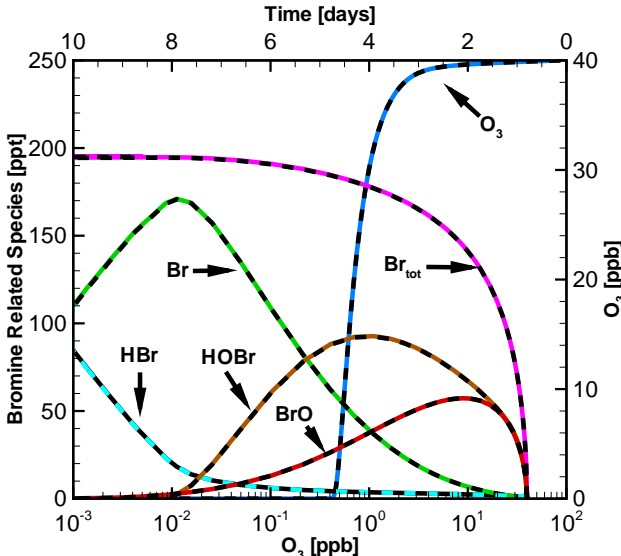

**Figure 4.** Change of ozone and principal bromine containing compounds plotted with an ozone coordinate axis and a time coordinate axis. The solid curves denote the simulation results obtained by using the original reaction mechanism, and the dashed curves represent the results for the reduced reaction mechanism which is obtained after the concentration sensitivity analysis. The curves of ozone mixing ratios use the axis of time while the principal bromine species correspond to the ozone coordinate axis. Note that the axis of time is in a reverse direction for clarity.





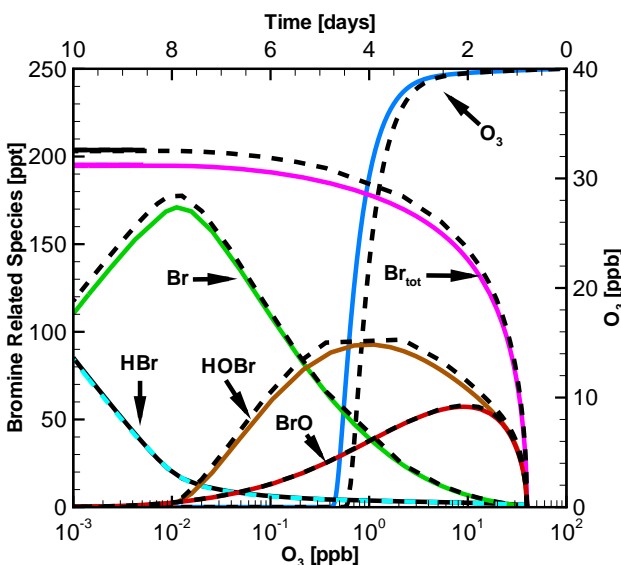

**Figure 5.** Change of ozone and principal bromine containing compounds plotted with an ozone coordinate axis and a time coordinate axis. The solid curves denote the simulation results obtained by using the original reaction mechanism, and the dashed curves represent the results for the reduced reaction mechanism which is obtained after the principal component analysis. The configuration of this figure is similar to that used in Fig. 4.





| | |
|---|---|
| (R1) | $O_3 + h\nu \rightarrow O(^1D) + O_2$ |
| (R2) | $O(^1D) + O_2 \rightarrow O_3$ |
| (R3) | $O(^1D) + N_2 \rightarrow O_3 + N_2$ |
| (R4) | $O(^1D) + H_2O \rightarrow 2OH$ |
| (R5) | $Br + O_3 \rightarrow BrO + O_2$ |
| (R6) | $Br_2 + h\nu \rightarrow 2Br$ |
| (R7) | $BrO + h\nu \xrightarrow{O_2} Br + O_3$ |
| (R8) | $BrO + BrO \rightarrow 2Br + O_2$ |
| (R9) | $BrO + BrO \rightarrow Br_2 + O_2$ |
| (R10) | $BrO + HO_2 \rightarrow HOBr + O_2$ |
| (R11) | $HOBr + h\nu \rightarrow Br + OH$ |
| (R12) | $CO + OH(+M) \xrightarrow{O_2} HO_2 + CO_2(+M)$ |
| (R13) | $Br + HO_2 \rightarrow HBr + O_2$ |
| (R14) | $HOBr + HBr \xrightarrow{aerosol} Br_2 + H_2O$ |
| (R15) | $HOBr + H^+ + Br^- \xrightarrow{ice} Br_2 + H_2O$ |
| (R16) | $Br + HCHO \xrightarrow{O_2} HBr + CO + HO_2$ |
| (R17) | $Br + CH_3CHO \xrightarrow{O_2} HBr + CH_3CO_3$ |
| (R18) | $Br_2 + OH \rightarrow HOBr + Br$ |
| (R19) | $HBr + OH \rightarrow H_2O + Br$ |
| (R20) | $Br + C_2H_2 \xrightarrow{3O_2} 2CO + 2HO_2 + Br$ |
| (R21) | $Br + C_2H_2 \xrightarrow{2O_2} 2CO + HO_2 + HBr$ |
| (R22) | $Br + C_2H_4 \xrightarrow{3.5O_2} 2CO + 2HO_2 + Br + H_2O$ |
| (R23) | $Br + C_2H_4 \xrightarrow{2.5O_2} 2CO + HO_2 + HBr + H_2O$ |
| (R24) | $CH_4 + OH \xrightarrow{O_2} CH_3O_2 + H_2O$ |
| (R25) | $BrO + CH_3O_2 \rightarrow Br + HCHO + HO_2$ |
| (R26) | $BrO + CH_3O_2 \rightarrow HOBr + HCHO + 0.5O_2$ |
| (R27) | $OH + O_3 \rightarrow HO_2 + O_2$ |
| (R28) | $OH + HO_2 \rightarrow H_2O + O_2$ |
| (R29) | $OH + H_2O_2 \rightarrow HO_2 + H_2O$ |
| (R30) | $OH + OH \xrightarrow{O_2} H_2O + O_3$ |
| (R31) | $HO_2 + O_3 \rightarrow OH + 2O_2$ |
| (R32) | $HO_2 + HO_2 \rightarrow O_2 + H_2O_2$ |
| (R33) | $C_2H_6 + OH \rightarrow C_2H_5 + H_2O$ |
| (R34) | $C_2H_5 + O_2 \rightarrow C_2H_4 + HO_2$ |
| (R35) | $C_2H_5 + O_2(+M) \rightarrow C_2H_5O_2(+M)$ |
| (R36) | $C_2H_4 + OH(+M) \xrightarrow{1.5O_2} CH_3O_2 + CO + H_2O(+M)$ |
| (R37) | $C_2H_4 + O_3 \rightarrow HCHO + CO + H_2O$ |
| (R38) | $C_2H_2 + OH(+M) \xrightarrow{1.5O_2} HCHO + CO + HO_2(+M)$ |
| (R39) | $C_3H_8 + OH \xrightarrow{2O_2} C_2H_5O_2 + CO + 2H_2O$ |
| (R40) | $HCHO + OH \xrightarrow{O_2} CO + H_2O + HO_2$ |
| (R41) | $CH_3CHO + OH \xrightarrow{O_2} CH_3CO_3 + H_2O$ |
| (R42) | $CH_3O_2 + HO_2 \rightarrow CH_3O_2H + O_2$ |
| (R43) | $CH_3O_2 + HO_2 \rightarrow HCHO + H_2O + O_2$ |
| (R44) | $CH_3OOH + OH \rightarrow CH_3O_2 + H_2O$ |
| (R45) | $CH_3OOH + OH \rightarrow HCHO + OH + H_2O$ |
| (R46) | $CH_3OOH + Br \rightarrow CH_3O_2 + HBr$ |

(a)

**Figure 6.** The importance of each reaction in the original reaction mechanism within different time periods, for (a) reactions (R1)-(R46), (b) reactions (R47)-(R92). Red contour denotes that this reaction is important through this time period, while blue contour represents that this reaction is identified as insignificant during this time stage. (Continued...)







(b)

**Figure 6.** The importance of each reaction in the original reaction mechanism within different time periods, for (a) reactions (R1)-(R46), (b) reactions (R47)-(R92). Red contour denotes that this reaction is important through this time period, while blue contour represents that this reaction is identified as insignificant during this time stage.