# Peer review of "Derivation of the Reduced Reaction Mechanisms of Ozone Depletion Events in the Arctic Spring by Using Concentration Sensitivity Analysis and Principal Component Analysis"

_Atmospheric Chemistry and Physics, 2016_

## Referee Comment (RC1) · Anonymous Referee #1 · 22 Sep 2016

By applying two mechanism reduction approaches in a photochemical box model, i.e., concentration sensitivity analysis and principal component analysis, Cao et al. simplified the complex chemical mechanism of the ozone depletion events (ODEs) in polar boundary layer, theoretically. Although the two methods are commonly used in ruling out unimportant reactions, they are first used to the ODEs as of my knowledge. For this I think this paper is worth publishing. Especially, the simulation results from the simplified reaction mechanisms are much agreeable with those from the original (complete) reaction mechanism, it is more convincing that this paper will serve as a useful reference for the theoretical analysis of ODEs. However, despite the good merit of

this paper there are some points worth exploring further. Therefore, I suggest revision before publishing in ACP. My concerns are listed below.

My major concern would be the limitation of the 0-D photochemical model used in the paper. As we know, chemical constituents are determined by both transport and chemical production/loss. Since the horizontal advection and the vertical mixing cannot be considered in the box model, I am worried about / interested in the role of transportation in redistributing ozone concentrations. The authors should also talk about the lifetime of ozone in polar boundary layer, in which we can roughly tell the importance of transport and chemical production/loss. This part could be jammed to the discussion part in section 4.

Another major complain is that the introduction is too long and it has less focus on the topic of this study. The authors used a lengthy context to introduce the historical finding of the ODEs while they really should have focused on is to introduce the efforts that have been taken in simplifying the reaction mechanism and why they decide to use the two methods towards ODEs. To me, the proper introduction starts since page 7, not page 2.

Other comments: P13, L2: It seems that fixed photolysis rates are applied to the mechanism. Then how is the photolysis frequencies actually specified in the model? Would the simulation result totally be different if the diurnal change of the photolysis rate is included? Please justify this as there is no convincing evidence in the present paper.

P13, L4: There are not enough details about the parameterization of the heterogeneous reactions in the mechanism. Those surface reactions are crucial to the depletion of ozone so detailed description could be helpful in understanding the bromine recycling processes. Apart from this, it will be helpful if the authors can discuss if the changes of the meteorological fields will affect their conclusions. For example, will the "redundant" reactions still be considered redundant if wind speed is increased?

Since the objective of this paper is to simply the reaction mechanism, I am wondering

if there are any quantitative measures of the numerical efficiency before and after the simplification?

Color bar is needed for Fig. 6 and Fig. 7. Although blue and red represent unimportant and important reactions, then what importance does green and yellow measures?

Supplements: reactions R14 and R15 in Table A1, what are the meanings of the parameters r, Dg, ... shown in the column of k? Please specify them in the caption of the table.

Comments on wording:

This paper could be tremendously shortened. Besides the much lengthy introduction that is less focused on the study, there are many redundant expressions (some used unnecessary clauses) to express an otherwise simple meaning. For example, • "The criterion shown in Eq. (5) means that" could be changed to "The criterion in Eq. (5) shows that"; • "components ... are thus obtained which are listed in Tab. 4" could be shortened as "components ... are listed in Tab. 4";

Also, in many clauses "which" should be replaced with "that".

---

## Referee Comment (RC2) · Anonymous Referee #2 · 22 Sep 2016

"Derivation of the Reduced Reaction Mechanism of Ozone Depletion Events in Polar Spring by Using Concentration Sensitivity Analysis and Principal Component Analysis," by Cao et al applies concentration sensitivity analysis (CSA) and principal component analysis (PCA) to a reaction scheme representative of the chemical reaction scheme that may govern Arctic ozone depletion events. CSA enables quantification of the dependence of species concentration on varied reaction rate constants while PCA flags unimportant reactions from a complex chemical scheme. These techniques also allows for the reduction of computational effort to complete simulations.

Through the CSA approach, they find that the original (92 reactions) and reduced reaction mechanism (81 reactions) both produce identical onsets of each time period of ODEs; simultaneously, the maximum deviation of bromine mixing ratios between the original and reduced reaction mechanisms is less than 1%. CSA also revealed 11 unimportant reactions. Applying PCA to the original reaction mechanism revealed the dependence on species on concentration on varied reaction rates and identified an additional 9 unimportant reactions (yielding 72 reactions for the reduced reaction mechanism from both CSA and PCA). PCA also revealed that the maximum deviation of the principal bromine species (Br, HBr, HOBr, BrO, and Brtot) between the original and reduced reaction mechanism was less than 10%. Overall, the original and reduced chemical mechanism for Arctic ODEs produced analogous model results via the box model KINAL.

I highly recommend this paper for publication. I enjoyed reading this manuscript given its robust and quantitative nature. Cao et al's aims to elucidate a fundamental query (what reaction mechanism quantitatively and explicitly governs polar tropospheric ODEs?) that still – that is, in the most explicit manner – via multiphase modeling – is still unanswered. This manuscript makes great strides at identifying (as a function of species concentration, reaction rates, and temporally) the dominant 72 reactions that may govern Arctic ODEs. This work is significant as it pinpoints the dominant reactions (in a multiphase fashion) that governs Arctic ODEs. The study is also important as it provides greater constraints on ice/snow reactions to investigate further in the lab/snowpack simulation chamber. I recommend that the authors refine the paper by emphasizing in a more transparent manner the key findings of the study – that is, identification of a concise reaction mechanism that may govern Arctic ODEs (from a more complex reaction mechanism) via a multiphase/box model approach, which in turn saves computational time. I can see this statistical approach becoming more valuable and applicable in the geosciences.

---

## Author Comment (AC2) · 25 Oct 2016

The authors thank the reviewer for the highly recommendation of this paper. In the following, we address the only issue raised by the reviewer:

*R2.1: I recommend that the authors refine the paper by emphasizing in a more transparent manner the key findings of the study – that is, identification of a concise reaction mechanism that may govern Arctic ODEs (from a more complex reaction mechanism) via a multiphase/box model approach, which in turn saves computational time.*

**A2.1:** Thanks a lot for the suggestion that the motivation of our present research should be discussed more clearly. Thus, we have improved the introduction section, especially explaining more why we did this study and how our findings can help to improve the current understanding of the ODE. Please see **lines 4-8 in page 6** of the revised manuscript.

---

## Author Response (AR1)

**Response to Reviewer #1**

The authors sincerely thank the reviewer for the valuable comments, which greatly contribute to an improvement of our paper.

In the following, we address the particular issues raised by the reviewer:

*R1.1: My major concern would be the limitation of the 0-D photochemical model used in the paper. As we know, chemical constituents are determined by both transport and chemical production/loss. Since the horizontal advection and the vertical mixing cannot be considered in the box model, I am worried about / interested in the role of transportation in redistributing ozone concentrations. The authors should also talk about the lifetime of ozone in polar boundary layer, in which we can roughly tell the importance of transport and chemical production/loss. This part could be jammed to the discussion part in section 4.*

**A1.1:** Thanks a lot for this comment. The reviewer is correct in saying that the ozone mixing ratio in the boundary layer is influenced by the joint effect of local chemistry, vertical mixing and horizontal advection. In our present model, the vertical turbulent mixing for redistributing chemical species in the 200 m boundary layer is implicitly accounted for by the estimation of the aerodynamic resistance in the computation of the heterogeneous reaction rates (see Eq. 18 in the revised manuscript, **line 25 in page 11**).

It is also known that the horizontal advection contributes to the occurrence and termination of ODEs observed at a fixed point to some extent. By investigating the origin of the ozone-depleted air at three observational sites in the springtime Arctic, Bottenheim and Chan (2006) revealed that the occurrence of ODEs is linked to a horizontal transport of the cold, ozone-depleted air across the Arctic Ocean covered with fresh sea ice. Their conclusion is also supported by the statistical analysis performed by Hirdman et al. (2009). Moreover, in a 3-D model study conducted by Toyota et al. (2011), it is found that the termination of ODEs during the Arctic spring is associated with enhanced boundary-layer wind transported from the south, carrying the ozone-rich air to the location under observation. However, by showing the dependency of the hourly-mean ozone level on the wind speed at Barrow, Alaska during the springtime of 2009, Helmig et al. (2012) suggested that ODEs are more frequently observed under a calm-wind condition (wind speed < 5 m s$^{-1}$). Thus, the relative importance of the horizontal advection for ODEs is still under debate. In order to clarify this, a fully coupled 3-D model is needed, which is beyond the limitations of the present box model. For a future development of the model used in this study, the horizontal advection process can be parameterized as a reaction sequence in the mechanism. After this parameterization, it is possible to implement the simplification approaches presented in this manuscript on the reaction mechanism with the inclusion of the horizontal advection.

A discussion about the role of horizontal advection in the depletion of ozone has been added in Sect. 4 as the reviewer suggested; please see **lines 9-22 in page 17** of the revised manuscript.

*R1.2: Another major complain is that the introduction is too long and it has less focus on the topic*

*of this study. The authors used a lengthy context to introduce the historical finding of the ODEs while they really should have focused on is to introduce the efforts that have been taken in simplifying the reaction mechanism and why they decide to use the two methods towards ODEs. To me, the proper introduction starts since page 7, not page 2.*

**A1.2:** Thanks. In the introduction section of the manuscript, we discussed a lot about previous numerical studies of ODEs since the simplification of the ODE mechanism outlined in this study is mostly helpful for the modeling work of ODEs. However, as the reviewer pointed out, the related context is too lengthy. Thus, we have shortened it significantly in the revised version of the paper; please see the related context **from line 4 in page 3 to line 14 in page 4** of the revised manuscript.

At present, to our knowledge, the efforts have been made to the simplification of the ODE reaction mechanism are still lacking except some preliminary work made by the author of this manuscript and co-workers (Cao and Gutheil, 2013; Cao et al., 2014). However, as the reviewer suggested, we added more content about why we did this study and how our findings can help to improve the current understanding of ODEs, see **lines 4-8 in page 6**.

*R1.3: P13, L2: It seems that fixed photolysis rates are applied to the mechanism. Then how is the photolysis frequencies actually specified in the model? Would the simulation result totally be different if the diurnal change of the photolysis rate is included? Please justify this as there is no convincing evidence in the present paper.*

**A1.3:** In our present model, we assumed a fixed value of solar zenith angle (SZA), 80° for the calculation of the photolysis frequencies. This fixed SZA value was also adopted by Lehrer et al. (2004) in their 1-D model study. Specifically, the photolysis frequencies, $J$ are evaluated by using a three-coefficient function (see Eq. 1 of this rebuttal) based on Röth's Anisotropic Radiation Transfer (ART) model (Röth 1992, 2002),

$$J = J_0 \exp(b[1 - \sec(c\chi)]) . \tag{1}$$

In Eq. (1), $\chi$ denotes the value of SZA. The coefficients $J_0$, $b$, $c$ are determined in the ART model under the conditions of SZA = 0°, 60° and 90°. The values of these parameters, $J_0$, $b$ and $c$, in the present model are listed in Tab. A1 of this rebuttal:

Tab. A1 Coefficients for the evaluation of the photolysis frequencies (Cao et al., 2016a).

| Species | $J_0$ [s$^{-1}$] | $b$ | $c$ | Species | $J_0$ [s$^{-1}$] | $b$ | $c$ |
|---|---|---|---|---|---|---|---|
| $O_3$ | $6.85 \times 10^{-5}$ | 3.510 | 0.820 | $CH_3O_2H$ | $1.60 \times 10^{-5}$ | 1.553 | 0.849 |
| $Br_2$ | $1.07 \times 10^{-1}$ | 0.734 | 0.900 | $C_2H_5O_2H$ | $1.60 \times 10^{-5}$ | 1.553 | 0.849 |
| $BrO$ | $1.27 \times 10^{-1}$ | 1.290 | 0.857 | $HNO_3$ | $1.39 \times 10^{-6}$ | 2.094 | 0.848 |
| $HOBr$ | $2.62 \times 10^{-3}$ | 1.216 | 0.861 | $NO_2$ | $2.62 \times 10^{-2}$ | 1.068 | 0.871 |
| $H_2O_2$ | $2.75 \times 10^{-5}$ | 1.595 | 0.848 | $NO_3 \rightarrow NO_2$ | $6.20 \times 10^{-1}$ | 0.608 | 0.915 |
| $HCHO \rightarrow HO_2$ | $1.03 \times 10^{-4}$ | 1.785 | 0.848 | $NO_3 \rightarrow NO$ | $7.03 \times 10^{-2}$ | 0.583 | 0.917 |
| $HCHO \rightarrow H_2$ | $1.08 \times 10^{-4}$ | 1.431 | 0.853 | $BrONO_2$ | $3.11 \times 10^{-3}$ | 1.270 | 0.859 |
| $C_2H_4O$ | $1.95 \times 10^{-5}$ | 4.050 | 0.710 | $BrNO_2$ | $1.11 \times 10^{-3}$ | 1.479 | 0.851 |

It should also be noted that during the evaluation of the parameters in the ART model, the sky is assumed cloud-free and the surface albedo is set equal to 1.0.

It has been proved in our earlier publications (Cao et al., 2014, 2016a) that the inclusion of the

SZA variation exerts only a minor influence on the temporal behavior of principal chemical species (see Fig. A1 of this rebuttal). This finding is also in consistence with the numerical investigation performed by Lehrer et al. (2004) for the impacts of SZA change on the rate of ozone loss. Thus, it indicates that the simulation results in the present manuscript with fixed value of SZA are capable of describing the main features of ODEs. The inclusion of the diurnal pattern will not significantly change the computational results as well as the major conclusions obtained in the present study.

[Figure]

Fig. A1 Evolution of the chemical species concentrations
with varying SZA in the 200 m boundary layer (Cao et al., 2016a).

As the model results with varying SZA have already been presented in our earlier publications (Cao et al., 2014, 2016a), in this manuscript, we only added the content about how the photolysis frequencies are estimated and a citation to our publications (Cao et al., 2014, 2016a) to illustrate that the variation of SZA makes little contribution to the change of the model results; please see **line 29 in page 10 to line 1 in page 11** of the revised manuscript.

*R1.4: P13, L4: There are not enough details about the parameterization of the heterogeneous reactions in the mechanism. Those surface reactions are crucial to the depletion of ozone so detailed description could be helpful in understanding the bromine recycling processes. Apart from this, it will be helpful if the authors can discuss if the changes of the meteorological fields will affect their conclusions. For example, will the "redundant" reactions still be considered redundant if wind speed is increased?*

**A1.4:** Thanks a lot for this comment. We have added the details about the parameterizations of the heterogeneous reactions occurring on the aerosols and ice/snow-covered surfaces in the revised manuscript. Please see **lines 8-26 in page 11** of the revised manuscript.

The reviewer also concerned about whether the changes of the local meteorological conditions would influence the conclusions of the present study or not. Obviously, modifications of the values of meteorological parameters such as the wind speed and the temperature will lead to the change of the reaction rates in the mechanism. As a result, a new reaction mechanism is

constructed and needs a complete analysis from the beginning. Thus, in order to clarify this concern, we conducted another simulation scenario by changing the wind speed from the original value 8 m s$^{-1}$ to 5 m s$^{-1}$ while other input parameters (e.g. boundary layer height) are kept the same. The redundant reactions indicated in the present study are still identified as unimportant in the concentration sensitivity analysis and the principal component analysis. Thus, we conclude that our conclusions achieved in this study are mostly valid for the typical polar condition.

It might be useful to inspect the range in which our conclusions are valid by performing a series of simulations under different meteorological conditions. A deeper and more thorough analysis of the computational results is also needed. We feel that this part could be the topic of another interesting paper. Therefore, we like to leave this simulation for a future publication and thus did not add the related content into the present manuscript. Thanks a lot for this valuable comment.

*R1.5: Since the objective of this paper is to simply the reaction mechanism, I am wondering if there are any quantitative measures of the numerical efficiency before and after the simplification.*
**A1.5:** Thanks a lot for this comment. We have estimated the computing time in our box model for different reaction mechanisms presented in this study. It is found that after the removal of the 11 redundant reactions indicated in the concentration sensitivity analysis, compared with the simulation with the implementation of the original reaction mechanism, the computing time decreases by 10%. In contrast to that, the reduced mechanism (39 species and 72 reactions) obtained after the principal component analysis causes an 18% drop of the computing time. Thus, it can be concluded that the simplification of the original reaction mechanism using these two approaches in the present study would significantly improve the numerical efficiency of the modeling work of ODEs. For a further improvement of the computational efficiency, it is needed to eliminate the redundant species from the mechanism, which has been discussed in the previous version of the manuscript (see the last paragraph of the paper). We have added the related content about how much the computational efficiency is improved in our box model in the revised manuscript; see **lines 28-34 in page 17**.

It should be noted that the improvement of the numerical efficiency depends on many factors such as the dimensionality of the numerical tool adopted and the available resources on the computation. Thus, the quantitative measure that how much the computing time is shortened outlined in this rebuttal only applies to our box model and the machine we used.

*R1.6: Color bar is needed for Fig. 6 and Fig. 7. Although blue and red represent unimportant and important reactions, then what importance does green and yellow measures?*
**A1.6:** The green and yellow contours in Figs. 6 and 7 of the previous version of the paper are generated due to the auto "smooth-out" function of the graphics software. We have re-drawn these two figures and added the color bars.

*R1.7: Supplements: reactions R14 and R15 in Table A1, what are the meanings of the parameters r, Dg . . . shown in the column of k? Please specify them in the caption of the table.*
**A1.7:** We have added a brief description of these parameters in the caption of Table S1 in the supplementary material.

*R1.8: This paper could be tremendously shortened. Besides the much lengthy introduction that is less focused on the study, there are many redundant expressions (some used unnecessary clauses) to express an otherwise simple meaning. For example, "The criterion shown in Eq. (5) means that" could be changed to "The criterion in Eq. (5) shows that"; "components ... are thus obtained which are listed in Tab. 4" could be shortened as "components ... are listed in Tab. 4"; Also, in many clauses "which" should be replaced with "that".*

**A1.8:** The authors greatly appreciate the suggestion of the reviewer for the wording improvement of the manuscript. We have revised the manuscript again and improved it. All the places where the reviewer pointed out have been corrected as well.

[revised manuscript text omitted]

|---|---|---|---|---|---|
| 1
$1.9\times10^6$ | (R15) | (R11) | (R20) | (R10) | (R5) |
| 2
$2.3\times10^5$ | (R20)
(R8) | (R5) | (R21) | (R7) | (R11) |
| 3
$3.4\times10^3$ | (R20)
(R75) | (R83)
(R11) | (R39) | (R15) | (R12) |
| 4
$2.6\times10^3$ | (R83)
(R82) | (R20)
(R8) | (R11)
(R41) | (R75) | (R24) |
| 5
$2.1\times10^3$ | (R5)
(R20) | (R21)
(R13) | (R7) | (R15) | (R39) |
| 6
$1.3\times10^3$ | (R41)
(R86) | (R12)
(R91) | (R21) | (R31) | (R83) |
| 7
$5.9\times10^2$ | (R21)
(R31) | (R5) | (R24) | (R12) | (R7) |
| 8
$4.6\times10^2$ | (R12)
(R19) | (R33) | (R10) | (R35) | (R41) |
| 9
$3.4\times10^2$ | (R70)
(R13) | (R55)
(R33) | (R19)
(R83) | (R69)
(R35) | (R62) |
| 10
$3.0\times10^2$ | (R35)
(R15) | (R33)
(R86) | (R10)
(R83) | (R24)
(R8) | (R17) |
| 11
$2.9\times10^2$ | (R24)
(R19) | (R89)
(R55) | (R35)
(R47) | (R70)
(R84) | (R86)
(R33) |
| 12
$2.7\times10^2$ | (R89)
(R23) | (R84) | (R70) | (R75) | (R10) |
| 13
$2.4\times10^2$ | (R86)
(R90) | (R21) | (R89) | (R24) | (R20) |
| 14
$2.0\times10^2$ | (R70)
(R24) | (R75)
(R3) | (R1)
(R53) | (R69) | (R51) |
| 15
$1.6\times10^2$ | (R14)
(R55) | (R1)
(R15) | (R51)
(R62) | (R3)
(R39) | (R35)
(R12) |
| 16
$1.4\times10^2$ | (R23)
(R53) | (R57)
(R86) | (R22)
(R42) | (R75) | (R6) |

| $\lambda$ | Dominant elements of the principal components | | | | |
|---|---|---|---|---|---|
| 17
$1.2\times10^2$ | (R53)
(R76) | (R10)
(R88) | (R51)
(R90) | (R17) | (R19) |
| 18
$1.1\times10^2$ | (R6)
(R3) | (R1)
(R55) | (R53)
(R13) | (R32)
(R76) | (R88) |
| 19
$9.7\times10^1$ | (R42) | (R88) | (R76) | (R17) | (R32) |
| 20
$8.8\times10^1$ | (R76)
(R57) | (R88)
(R17) | (R51) | (R42) | (R14) |
| 21
$8.3\times10^1$ | (R6)
(R51) | (R14)
(R23) | (R32)
(R47) | (R19)
(R75) | (R55) |
| 22
$7.5\times10^1$ | (R17)
(R57) | (R42)
(R55) | (R88)
(R6) | (R11) | (R76) |
| 23
$6.9\times10^1$ | (R6)
(R3) | (R1) | (R14) | (R75) | (R26) |
| 24
$5.9\times10^1$ | (R31) | (R10) | (R39) | (R23) | (R11) |
| 25
$5.5\times10^1$ | (R53)
(R39) | (R62)
(R17) | (R51) | (R55) | (R26) |
| 26
$4.9\times10^1$ | (R23)
(R62) | (R58)
(R46) | (R59) | (R13) | (R90) |
| 27
$4.0\times10^1$ | (R90)
(R53) | (R57)
(R24) | (R13)
(R14) | (R55) | (R47) |
| 28
$3.7\times10^1$ | (R34)
(R13) | (R46)
(R62) | (R26) | (R8) | (R90) |
| 29
$3.4\times10^1$ | (R53)
(R12) | (R47)
(R24) | (R58) | (R34) | (R26) |
| 30
$3.0\times10^1$ | (R90) | (R34) | (R57) | (R26) | |
| 31
$2.8\times10^1$ | (R34)
(R26) | (R11) | (R13) | (R14) | (R78) |
| 32
$2.5\times10^1$ | (R13)
(R31) | (R58) | (R50) | (R55) | (R59) |

[revised manuscript text omitted]

$$
\begin{array}{ll}
(R47) & CH_3O_2 + CH_3O_2 \rightarrow CH_3OH + HCHO + O_2 \\
(R48) & CH_3O_2 + CH_3O_2 + O_2 \rightarrow 2HCHO + 2HO_2 \\
(R49) & CH_3OH + OH + O_2 \rightarrow HCHO + HO_2 + H_2O \\
(R50) & C_2H_5O_2 + C_2H_5O_2 \rightarrow C_2H_5O + C_2H_5O + O_2 \\
(R51) & C_2H_5O + O_2 \rightarrow CH_3CHO + HO_2 \\
(R52) & C_2H_5O + O_2 \rightarrow CH_3O_2 + HCHO \\
(R53) & C_2H_5O_2 + HO_2 \rightarrow C_2H_5OOH + O_2 \\
(R54) & C_2H_5OOH + OH \rightarrow C_2H_5O_2 + H_2O \\
(R55) & C_2H_5OOH + Br \rightarrow C_2H_5O_2 + HBr \\
(R56) & OH + OH(+M) \longrightarrow H_2O_2(+M) \\
(R57) & H_2O_2 + h\nu \rightarrow 2OH \\
(R58) & HCHO + h\nu + 2O_2 \rightarrow 2HO_2 + CO \\
(R59) & HCHO + h\nu \rightarrow H_2 + CO \\
(R60) & C_2H_4O + h\nu \rightarrow CH_3O_2 + CO + +HO_2 \\
(R61) & CH_3O_2H + h\nu \rightarrow OH + HCHO + HO_2 \\
(R62) & C_2H_5O_2H + h\nu \rightarrow C_2H_5O + OH \\
(R63) & NO + O_3 \rightarrow NO_2 + O_2 \\
(R64) & NO + HO_2 \rightarrow NO_2 + OH \\
(R65) & NO_2 + O_3 \rightarrow NO_3 + O_2 \\
(R66) & NO_2 + OH(+M) \rightarrow HNO_3(+M) \\
(R67) & NO + NO_3 \rightarrow 2NO_2 \\
(R68) & HONO + OH \rightarrow NO_2 + H_2O \\
(R69) & HO_2 + NO_2(+M) \rightarrow HNO_4(+M) \\
(R70) & HNO_4(+M) \rightarrow NO_2 + HO_2(+M) \\
(R71) & HNO_4 + OH \rightarrow NO_2 + H_2O + O_2 \\
(R72) & NO + OH(+M) \rightarrow HONO(+M) \\
(R73) & OH + NO_3 \rightarrow NO_2 + HO_2 \\
(R74) & HNO_3 + h\nu \rightarrow NO_2 + OH \\
(R75) & NO_2 + h\nu + O_2 \rightarrow NO + O_3 \\
(R76) & NO_3 + h\nu + O_2 \rightarrow NO_2 + O_3 \\
(R77) & NO_3 + h\nu \rightarrow NO + O_2 \\
(R78) & NO + CH_3O_2 + O_2 \rightarrow HCHO + HO_2 + NO_2 \\
(R79) & NO_3 + CH_3OH + O_2 \rightarrow HCHO + HO_2 + HNO_3 \\
(R80) & NO_3 + HCHO + O_2 \rightarrow CO + HO_2 + HNO_3 \\
(R81) & NO + C_2H_5O_2 + O_2 \rightarrow CH_3CHO + NO_2 + HO_2 \\
(R82) & NO + CH_3CO_3 + O_2 \rightarrow CH_3O_2 + NO_2 + CO_2 \\
(R83) & NO_2 + CH_3CO_3(+M) \rightarrow PAN(+M) \\
(R84) & Br + NO_2(+M) \rightarrow BrNO_2(+M) \\
(R85) & Br + NO_3 \rightarrow BrO + NO_2 \\
(R86) & BrO + NO_2(+M) \rightarrow BrONO_2(+M) \\
(R87) & BrO + NO \rightarrow Br + NO_2 \\
(R88) & BrONO_2 + h\nu \rightarrow NO_2 + BrO \\
(R89) & BrNO_2 + h\nu \rightarrow NO_2 + Br \\
(R90) & BrONO_2 + H_2O \xrightarrow{aerosol} HOBr + HNO_3 \\
(R91) & PAN + h\nu \rightarrow NO_2 + CH_3CO_3 \\
(R92) & BrONO_2 + H_2O \xrightarrow{ice} HOBr + HNO_3 \\
\end{array}
$$

(b)

**Figure 6.** The importance of each reaction in the original reaction mechanism within different time periods, for (a) reactions (R1)-(R46), (b) reactions (R47)-(R92). Yellow contour denotes that this reaction is important through this time period, while black contour represents that this reaction is identified as insignificant during this time stage.